# ADVERSARIAL UNLEARNING OF POISONED FEATURES FOR BACKDOOR DEFENSE IN FEDERATED LEARNING

## ABSTRACT

Federated learning (FL) enables collaborative model training without exposing local data but is highly vulnerable to backdoor attacks, particularly under non-IID client distributions and persistent malicious participants. Existing defenses often rely on robust aggregation or auxiliary data, yet their effectiveness diminishes under challenging conditions such as low poisoning ratios and heterogeneous data, and they remain susceptible to adaptive or stealthy adversaries. We propose *adversarial unlearning of poisoned features* (AUPF), an in-training defense that generates adversarial perturbations on benign clients to expose vulnerable decision boundaries and explicitly regularizes the feature representations of clean and perturbed samples. This feature-level alignment suppresses poisoned associations and ensures that robustness acquired locally propagates to the global model despite dynamic updates and client heterogeneity. We design a bi-level optimization framework that integrates seamlessly with FL training and show that it achieves computational efficiency comparable to lightweight baselines while avoiding the scalability issues of prior defenses. Extensive experiments across diverse datasets, attack strategies, and non-IID scenarios demonstrate that AUPF consistently achieves lower attack success rates while maintaining high clean accuracy, establishing it as an effective and scalable defense for backdoor-resilient federated learning.

## 1 INTRODUCTION

Federated learning (FL) is a distributed learning paradigm designed to mitigate privacy concerns inherent in centralized machine learning (McMahan et al., 2017; Kairouz et al., 2021). It enables a large number of clients to collaboratively train a global model without exposing their private local data. However, previous studies have shown that federated learning is inherently vulnerable to backdoor attacks, primarily due to its decentralized training paradigm and restricted access to local training data (Bagdasaryan et al., 2020; Fang & Chen, 2023). In particular, malicious clients can inject backdoors into the global model by manipulating their local training data during the collaborative training process, causing the model to behave normally on clean inputs while misclassifying any input containing a predefined trigger (*i.e.*, poisoned input) into a target class (Xie et al., 2020; Zhang et al., 2022b; 2023a; Lyu et al., 2023).

To address the challenge of backdoor attacks in FL, extensive research has focused on developing training algorithms that make models inherently robust to such threats. However, current federated learning backdoor defense methods typically suffer from one or more of the following limitations.

- *Many defenses struggle to withstand continuous attacks*, in which malicious clients launch attacks in every communication round from the very beginning of training (Rieger et al., 2022; Zhang et al., 2023b). Specifically, these defenses often assume that malicious clients are only active for a limited number of rounds or at a low frequency (Bagdasaryan et al., 2020). For instance, FLIP assumes the attack is launched only after the global model has converged (Zhang et al., 2023b). In the most extreme case, post-training defenses assume the model has already been poisoned and focus on removing the backdoor without any further adversarial involvement (Li et al., 2021; Zeng et al., 2022; Zheng et al., 2022; Xu et al., 2024). In contrast, a more desirable yet considerably more challenging objective is to develop a model that is inherently robust to backdoor attacks during training, particularly in the context of federated learning.
- *Defense mechanisms that depend on the cooperation of clients are often ineffective against adaptive attacks*, as malicious clients may intentionally deviate from or ignore the prescribed defense

strategies (Huang et al., 2024; 2023b). Table S1 in the Appendix illustrates adaptive attacks on the SOTA defenses FDCR (Huang et al., 2024) and Lockdown (Huang et al., 2023b).

- Robust aggregation-based defenses that rely on distinguishing between the updates of benign and malicious clients are *sensitive to data heterogeneity across clients or poison ratios used by malicious clients (Pillutla et al., 2022; Blanchard et al., 2017; Qin et al., 2024; Huang et al., 2024)*. As shown in Figures 2 and 3, these defenses exhibit significant performance degradation under *high data heterogeneity* and *low poison ratios*, respectively, on the CIFAR-10 dataset. This vulnerability arises because, in highly non-IID settings or when the poison ratio is low, the differences between benign and malicious updates become increasingly subtle, making it more difficult for the server to reliably identify adversarial clients. Notably, low poison ratios are often deliberately used by attackers to enhance the stealthiness of backdoor attacks. Furthermore, the effectiveness of such defenses also depends on the overall proportion of malicious clients in the system.

- The effectiveness of defenses *relying on auxiliary datasets* is inherently constrained by their availability and quality (Cao et al., 2021a; Park et al., 2021). Specifically, federated learning is designed to preserve data privacy by keeping data localized on client devices, which makes it challenging for the server to obtain appropriate auxiliary datasets. Moreover, even when public datasets are accessible, the performance of these defenses is highly sensitive to the similarity between the auxiliary data and the original training data.

In summary, practical defenses for federated learning must address four key challenges: 1) continuous attacks, 2) adaptive adversaries, 3) heterogeneous client data and low poison ratios, and 4) the absence of auxiliary datasets. To this end, we propose adversarial unlearning of poisoned features (AUPF), a client-side defense that continuously mitigates backdoors under persistent malicious participation and withstands adaptive adversaries through adversarial unlearning with universal perturbations. It also preserves robustness under heterogeneous data and low poison ratios by relying on standard aggregation, which avoids the pitfalls of robust aggregation under non-IID settings or low poison ratios. Finally, AUPF requires no auxiliary datasets, as it leverages only clients' own data and models.

The core idea of AUPF is to perform adversarial unlearning during federated training. Each benign client generates adversarial examples using a universal perturbation $\Delta$, which is optimized not to reconstruct a specific trigger but to expose vulnerable directions in the decision boundary. Since backdoor triggers exploit such vulnerabilities, training against $\Delta$ enhances robustness without requiring knowledge of the attacker's design. Starting from the global model $\mathbf{w}^t$, which aggregates updates from both benign and malicious clients, each local model $\mathbf{w}_i^t$ may inherit poisoned features. By applying adversarial unlearning locally, clients remove these associations while preserving task accuracy, and the resulting robustness propagates to the global model through aggregation, ensuring continuous mitigation throughout training.

This work makes the following key contributions:

- **Adversarial unlearning of poisoned features.** We propose AUPF, an in-training defense that goes beyond prediction-loss minimization by explicitly aligning clean and adversarial representations. This feature-level unlearning removes poisoned associations and propagates local robustness to the global model.

- **Federated-specific and efficient design.** AUPF directly addresses challenges unique to federated learning, including dynamic global updates and non-IID data, without relying on auxiliary datasets or trigger knowledge. We design a bi-level optimization framework that integrates seamlessly with FL training, incurring overhead comparable to FedAvg and Lockdown while avoiding the $O(C^2)$ complexity of FLIP, thereby scaling more efficiently (see Table 3).

- **Comprehensive empirical validation.** We evaluate AUPF on multiple datasets against four advanced backdoor attacks and eleven representative defenses. AUPF consistently achieves low attack success rates while maintaining high clean accuracy across diverse attack types, poisoning ratios, attacker fractions, and client scales, with particularly strong performance under challenging non-IID settings.

## 1.1 RELATED WORK

Existing defense strategies against backdoor attacks in FL can be broadly categorized into four classes. 1) Robust aggregation-based defenses (Özdayi et al., 2021; Zhang et al., 2022a; Pillutla et al., 2022) attempt to detect and exclude malicious clients during model aggregation, operating under the assumption that benign updates are relatively similar, whereas malicious ones deviate significantly due to their backdoor objectives. However, these methods often suffer performance degradation under

high data heterogeneity across clients or lower poison ratio. 2) Certified robustness-based defenses (Xie et al., 2021; Cao et al., 2021b; Panda et al., 2022) offer theoretical guarantees against backdoor attacks, but typically rely on strong assumptions, such as bounded backdoor strength or a limited number of malicious clients, which may not be realistic in practice. 3) Adversarial training-based defenses (Zizzo et al., 2020; Shah et al., 2021; Chen et al., 2022; Weng et al., 2020; Wei et al., 2023) seek to improve global model robustness by applying adversarial training on local clients before model aggregation. While these methods strengthen robustness against adversarial examples, they are not specifically designed to counter targeted backdoor attacks and thus offer limited protection against such threats. To defend against backdoor attacks, (Zhang et al., 2023b) propose FLIP, a method that generates adversarial samples by attempting to invert the backdoor trigger on benign clients. However, the computational cost of their trigger inversion process scales quadratically with the number of class labels, making it impractical for datasets with a large class count. 4) Auxiliary dataset-based defenses (Park et al., 2021; Sturluson et al., 2021; Cao et al., 2021a; Lee et al., 2024; Wu et al., 2024) are typically deployed on the server and rely on access to a small trusted validation dataset. This auxiliary data is often used to detect malicious client updates by evaluating their consistency with expected behavior. Consequently, these methods implicitly assume that malicious clients exhibit behavior distinguishable from benign ones based on certain indicators measured on the auxiliary dataset, while this assumption may not hold in practice.

## 2 PROBLEM SETUP

**Threat Model.** We consider a federated learning system in which a subset of clients may be compromised. These malicious clients aim to implant a backdoor into the global model by poisoning their local training data. Specifically, they train on inputs modified with a predefined trigger pattern $\boldsymbol{\delta}$, such that any clean input $\mathbf{x}$ containing $\boldsymbol{\delta}$, represented as the poisoned input $\mathbf{x} \oplus \boldsymbol{\delta}$, is misclassified as a target label $\widetilde{y}$, rather than its true label $y$. We define the *poison ratio* as the fraction of poisoned samples within the local training dataset. To ensure stealthiness, the attack must also preserve the model's performance on clean inputs, with no significant degradation in main task accuracy.

To model adaptive attacks, we assume that adversaries have full control over their local training data and procedures, allowing them to arbitrarily modify both to evade or circumvent prescribed defenses. In line with standard federated learning assumptions, malicious clients have no access to additional information from the server or other clients. We further adopt a strong threat model in which malicious clients can launch attacks in any communication round and at arbitrary frequencies—potentially engaging in continuous attacks. In the worst-case scenario, they participate and attack persistently throughout the entire training process. We assume that benign clients have no knowledge of the trigger pattern $\boldsymbol{\delta}$ or the target label $\widetilde{y}$. The global server performs standard aggregation (e.g., FedAvg (McMahan et al., 2017)) and does not rely on any auxiliary server-side data.

**Defense Goal.** Our objective is to design a robust local training algorithm for federated learning that enables the global model to achieve high performance on the main task (clean inputs) while effectively defending against backdoor attacks (poisoned inputs), even in the presence of malicious clients. The defense must operate without relying on auxiliary server-side data, or prior knowledge of the attack strategy (*e.g.*, trigger pattern and target label), and should remain fully compatible with the standard federated learning paradigm.

**Notations.** We denote the global model of federated learning at round $t$ by $\mathbf{w}^t$, and the local model of the $i$-th client at the same round by $\mathbf{w}_i^t$. The deep neural network $f$, parameterized by $\mathbf{w}^t$, is

$$f_{\mathbf{w}^t}(\mathbf{x}) := f_{\mathbf{w}^t}^c \circ f_{\mathbf{w}^t}^e(\mathbf{x}), \tag{1}$$

where $f_{\mathbf{w}^t}^e$ denotes the feature extractor (*i.e.*, the network excluding the final fully connected layer), which maps the input $\mathbf{x}$ into a high-level feature representation. The component $f_{\mathbf{w}^t}^c$ denotes the final fully connected layer of the model, which transforms the extracted features into the model's prediction output.

We denote the local training dataset of the $i$-th client by $\mathcal{D}_i$, where the $j$-th sample is represented as $(\mathbf{x}_j^i, y_j^i)$. If the $i$-th client is *malicious*, its local dataset is assumed to consist of two disjoint subsets: clean data, denoted by $\widehat{\mathcal{D}}_i$, and poisoned data, denoted by $\widetilde{\mathcal{D}}_i$, such that $\mathcal{D}_i = \widehat{\mathcal{D}}_i \cup \widetilde{\mathcal{D}}_i$. For a malicious client, each poisoned sample is denoted by $(\widetilde{\mathbf{x}}_j^i, \widetilde{y})$, where the poisoned input $\widetilde{\mathbf{x}}_j^i := \mathbf{x}_j^i \oplus \boldsymbol{\delta}$ is obtained by perturbing the original input $\mathbf{x}_i^j$ with the pre-defined trigger pattern $\boldsymbol{\delta}$. Then its local

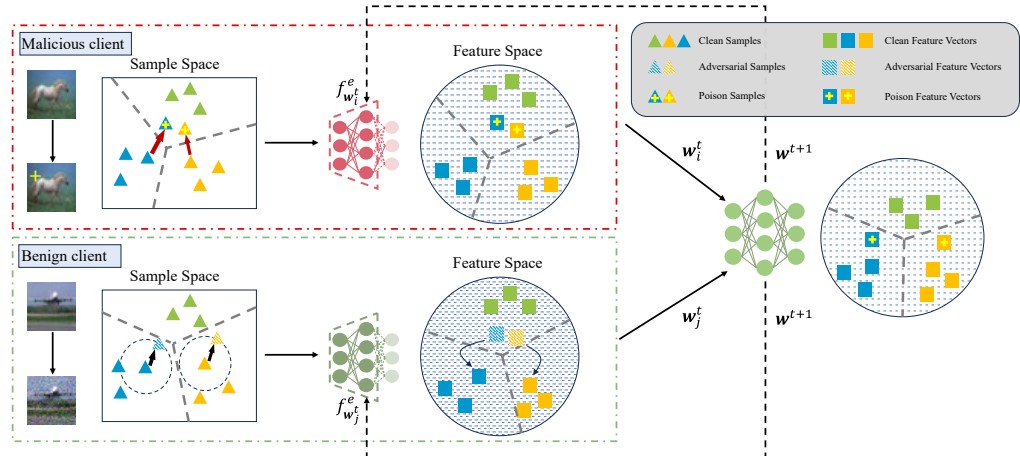

Figure 1: Overview of the proposed adversarial unlearning of poisoned features (AUPF). The top-left (red box) illustrates malicious clients launching a backdoor attack by adding a trigger to clean samples and relabeling them with the attacker-specified target. The bottom-left (green box) shows benign clients crafting adversarial examples to expose fragile decision boundaries and aligning clean and perturbed inputs to adversarially unlearn poisoned features, thereby hardening these vulnerable directions. After local training, all clients upload their updates to the server, which performs standard aggregation without distinguishing between benign and malicious clients. As illustrated on the right, AUPF mitigates backdoor effects at the global level, where poisoned samples are mapped back toward their original feature space, even under standard aggregation.

objective is defined as

$$\min_{\mathbf{w}} \left\{ \frac{1}{|\widehat{\mathcal{D}}_i|} \sum_{(\mathbf{x}_j^i, y_j^i) \in \widehat{\mathcal{D}}_i} L\big(f_{\mathbf{w}}\big(\mathbf{x}_j^i\big), y_j^i\big) + \frac{1}{|\widetilde{\mathcal{D}}_i|} \sum_{(\widetilde{\mathbf{x}}_j^i, \widetilde{y}) \in \widetilde{\mathcal{D}}_i} L\big(f_{\mathbf{w}}\big(\widetilde{\mathbf{x}}_j^i\big), \widetilde{y}\big) \right\}, \qquad (2)$$

where $L$ is the loss function.

## 3 ADVERSARIAL UNLEARNING OF POISONED FEATURES

Prior work, such as Neural Cleanse (Wang et al., 2019), has shown that backdoor triggers create shortcuts in the decision space by mapping inputs from the region of a source class to that of an attacker-specified target class. This attack mechanism exploits inherent vulnerabilities in the model's decision boundary. In federated learning, malicious clients leverage this property by injecting poisoned data locally and uploading poisoned updates, thereby coercing the global model into internalizing spurious associations. Formally, a backdoor attack transforms a clean sample $(\mathbf{x}_j^i, y_j^i)$ into a poisoned sample $(\mathbf{x}_j^i \oplus \boldsymbol{\delta}, \widetilde{y})$, where $\boldsymbol{\delta}$ is the trigger pattern and $\widetilde{y}$ is the attacker-chosen target label. Intuitively, these poisoned samples shift the decision boundary toward fragile directions in the feature space, creating adversarial shortcuts that allow the trigger $\boldsymbol{\delta}$ to override the original semantics of $\mathbf{x}_j^i$. Once such poisoned updates are aggregated, the vulnerabilities become embedded in the global model and can be reliably exploited by the attacker at inference time.

To address this inherent vulnerability, we propose **adversarial unlearning of poisoned features (AUPF)**. The key idea is that adversarial examples can be crafted to expose fragile decision boundaries, and training the model to resist them enables it to unlearn backdoor-induced associations. Fig. 1 presents an overview of AUPF. Formally, we learn a universal perturbation $\boldsymbol{\Delta}$ that, when added to all local training samples, maximizes the training loss and drives the model toward incorrect predictions. Given the local model $\mathbf{w}_i^t$ of the $i$-th benign client at round $t$, the perturbation is obtained as:

$$\boldsymbol{\Delta} = \arg\max_{\|\boldsymbol{\Delta}\| \leq \rho} \frac{1}{|\mathcal{D}_i|} \sum_{(\mathbf{x}_j^i, y_j^i) \in \mathcal{D}_i} L\big(f_{\mathbf{w}_i^t}\big(\mathbf{x}_j^i \oplus \boldsymbol{\Delta}\big), y_j^i\big), \qquad (3)$$

where $\rho$ bounds the perturbation strength. Unlike sample-specific perturbations that only capture vulnerabilities of individual inputs, a universal perturbation exposes common fragile directions shared across the data distribution. Crucially, the universal perturbation in AUPF is *not intended to reconstruct or approximate the attacker's trigger*; instead, it aims to expose and harden these vulnerable directions, thereby suppressing poisoned features and improving robustness against a broad class of potential triggers.

Although universal perturbations are computed locally on benign clients, they remain effective against global backdoors. This is because each benign client initializes its local model $\mathbf{w}_i^t$ from the global model $\mathbf{w}^t$, which already aggregates updates from both benign and malicious clients. As a result, $\mathbf{w}_i^t$ inherits not only the general decision structure of $\mathbf{w}^t$ but also its backdoor-related vulnerabilities. When universal perturbations are optimized on benign data with respect to $\mathbf{w}_i^t$, they naturally probe these vulnerabilities, since the local model closely approximates the global model. Thus, adversarial unlearning on benign clients directly targets weaknesses in the global model, and the robustness gained locally is propagated upward through aggregation.

Once $\mathbf{\Delta}$ is obtained, we incorporate the perturbed samples $(\mathbf{x}_j^i \oplus \mathbf{\Delta}, y_j^i)$ into the local training process to facilitate backdoor unlearning. We construct the adversarial dataset $\{ (\mathbf{x}_j^i \oplus \mathbf{\Delta}, y_j^i) \mid (\mathbf{x}_j^i, y_j^i) \in \mathcal{D}_i \}$ and use it to optimize the following training objective for the $i$-th benign client:

$$\mathbf{w}_i^t = \arg\min_{\mathbf{w}} \left\{ \frac{1}{|\mathcal{D}_i|} \sum_{(\mathbf{x}_j^i, y_j^i) \in \mathcal{D}_i} L(f_{\mathbf{w}}(\mathbf{x}_j^i), y_j^i) + \frac{\lambda_1}{|\mathcal{D}_i|} \sum_{(\mathbf{x}_j^i, y_j^i) \in \mathcal{D}_i} L(f_{\mathbf{w}}(\mathbf{x}_j^i \oplus \mathbf{\Delta}), y_j^i) \right\}, \qquad (4)$$

where the second term encourages the model to correctly classify adversarially perturbed inputs as their true labels $y_j^i$, thereby reducing sensitivity to potential backdoor triggers and suppressing the backdoor effect. The hyperparameter $\lambda_1 > 0$ controls the strength of adversarial unlearning. By integrating adversarial sample generation into local training, the optimization problem for the $i$-th benign client can be formulated as a bi-level objective:

$$\mathbf{w}_i^t = \arg\min_{\mathbf{w}} \left\{ \frac{1}{|\mathcal{D}_i|} \sum_{(\mathbf{x}_j^i, y_j^i) \in \mathcal{D}_i} L(f_{\mathbf{w}}(\mathbf{x}_j^i), y_j^i) + \frac{\lambda_1}{|\mathcal{D}_i|} \sum_{(\mathbf{x}_j^i, y_j^i) \in \mathcal{D}_i} L(f_{\mathbf{w}}(\mathbf{x}_j^i \oplus \mathbf{\Delta}), y_j^i) \right\} \qquad (5)$$

$$\text{s.t. } \mathbf{\Delta} = \arg\max_{\|\mathbf{\Delta}\| \leq \rho} \frac{1}{|\mathcal{D}_i|} \sum_{(\mathbf{x}_j^i, y_j^i) \in \mathcal{D}_i} L(f_{\mathbf{w}_i^t}(\mathbf{x}_j^i \oplus \mathbf{\Delta}), y_j^i),$$

where the inner maximization seeks a universal perturbation $\mathbf{\Delta}$ that maximally degrades performance on benign data, while the outer minimization updates the local model parameters $\mathbf{w}_i^t$ to preserve task accuracy and enhance resilience against backdoor attacks by jointly minimizing standard and adversarial losses.

In federated learning, applying Eq. (5) mitigates backdoor effects within benign clients' local models and data, yielding a clean global model in a single round. However, the global model evolves dynamically in federated learning and may deviate significantly from individual clients' local models, particularly under non-IID data distributions. These characteristics introduce two major challenges for federated adversarial training: 1) **Dynamic global model.** Unlike centralized adversarial training, where perturbations are optimized against a static model, perturbations in federated learning are generated with respect to local models while the global model keeps changing. This evolution complicates defense: even if adversarial training suppresses backdoors locally, malicious clients can continually reintroduce them into the global model through repeated participation and aggregation. 2) **Non-IID local data.** Centralized adversarial training benefits from access to the full dataset when generating perturbations. In contrast, federated learning restricts perturbation generation to each client's local data, and the inherent non-IID nature of these distributions further hinders the transfer of locally learned robustness to the global model.

To address these challenges, we propose adversarial unlearning of poisoned features (AUPF). Unlike prior defenses that mitigate backdoors solely by minimizing the prediction loss of adversarial examples, AUPF also enforces feature-level consistency between clean and adversarial inputs. This motivation stems from the observation that, in non-IID federated learning, feature representations are more transferable across clients, whereas the classification layer remains highly sensitive to local data distributions. Similar insights appear in personalized federated learning, where a shared feature extractor combined with client-specific classification heads is a common strategy (Collins et al., 2021; Oh et al., 2022; Mclaughlin & Su, 2024). Specifically, we introduce an additional regularization term to align clean and adversarial features:

$$\mathbf{w}_i^t = \arg\min_{\mathbf{w}} \left\{ \frac{1}{|\mathcal{D}_i|} \sum_{(\mathbf{x}_j^i, y_j^i) \in \mathcal{D}_i} L(f_{\mathbf{w}}(\mathbf{x}_j^i), y_j^i) + \frac{\lambda_1}{|\mathcal{D}_i|} \sum_{(\mathbf{x}_j^i, y_j^i) \in \mathcal{D}_i} L(f_{\mathbf{w}}(\mathbf{x}_j^i \oplus \mathbf{\Delta}), y_j^i) + \frac{\lambda_2}{|\mathcal{D}_i|} \sum_{(\mathbf{x}_j^i, y_j^i) \in \mathcal{D}_i} \|f_{\mathbf{w}}^e(\mathbf{x}_j^i) - f_{\mathbf{w}}^e(\mathbf{x}_j^i \oplus \mathbf{\Delta})\|^2 \right\}$$

$$(6)$$

$$\text{s.t. } \mathbf{\Delta} = \arg\max_{\|\mathbf{\Delta}\| \leq \rho} \frac{1}{|\mathcal{D}_i|} \sum_{(\mathbf{x}_j^i, y_j^i) \in \mathcal{D}_i} L(f_{\mathbf{w}_i^t}(\mathbf{x}_j^i \oplus \mathbf{\Delta}), y_j^i),$$

where $f_{\mathbf{w}}^e$ denotes the feature extractor. This regularization enforces similarity between clean and adversarial features, yielding a more stable and robust defense.

Embedding robustness at the representation level enables defenses to withstand both the *dynamics of global updates* and *heterogeneous client distributions*. For the former, robustness anchored in the shared extractor persists across communication rounds: as shown in the ASR trend (Fig. S5, Appendix Sec. A.6), without feature unlearning ASR fluctuates sharply, whereas with feature unlearning it drops significantly and remains stable. For the latter, feature-level consistency alleviates the drift introduced by non-IID data: in t-SNE visualizations (Fig. S6, Appendix Sec. A.6), under FedAvg poisoned samples form a separate cluster, whereas with AUPF they realign with their original class clusters. Together, these results demonstrate that adversarial feature unlearning embeds robustness directly into the global representation space, ensuring a stronger and more consistent defense.

**Local training for malicious clients.** In FLIP (Zhang et al., 2023b), malicious clients are assumed to follow standard poisoned training, as in problem Eq. (2). This assumption is idealized, since in practice attackers may adapt once defenses are deployed. We therefore consider an *adaptive attack* where malicious clients apply adversarial training on their *poisoned samples* $\widetilde{\mathcal{D}}_i$, explicitly hardening the backdoor against unlearning (see Appendix Sec. B.1). Their procedure mirrors that of benign clients but restricts adversarial perturbations to poisoned data, thereby reinforcing the malicious signal. We term this strategy adversarial hardening of poisoned features. Our experiments show that AUPF remains robust even under such adaptive attacks: benign clients continually estimate perturbations from local data, which enables them to unlearn residual backdoor influence and counteract its reintroduction into the global model.

**Discussion of Novelty and Distinction from Prior Adversarial Unlearning Methods.** To clarify the main novelty of AUPF, we discuss its differences from prior adversarial training in Appendix C. In addition, we summarize the key distinctions between AUPF and the approaches proposed in (Wei et al., 2023) and (Zeng et al., 2022) in Appendix C.1. Overall, these discussions further clarify the conceptual and algorithmic novelty of AUPF.

## 4 EXPERIMENTS

In this section, we empirically evaluate the effectiveness of the proposed AUPF by comparing it against several state-of-the-art defense methods under a variety of attack strategies.

### 4.1 EXPERIMENT SETUP

**Datasets and Models.** We evaluate our method on three image classification benchmarks: CIFAR-10, CIFAR-100 (Krizhevsky, 2009) and FashionMNIST (Xiao et al., 2017). Following prior works on federated backdoor attacks (Xie et al., 2020; Zhang et al., 2023a; Zhuang et al., 2024) and defenses (Huang et al., 2023b; 2024), we adopt ResNet-18 (He et al., 2016) as the backbone for CIFAR-10 and CIFAR-100. ResNet-18 has become the de facto standard in this line of research, facilitating fair comparisons across both attack and defense methods. For FashionMNIST, due to its lower complexity, we employ a lightweight four-layer CNN.

**Federated Learning Setup.** By default, we set the number of clients to $N = 40$. To simulate non-IID data distributions across clients, we partition the data using a Dirichlet distribution with concentration parameter $\alpha = 0.5$. We further evaluate the sensitivity of the proposed defense method to data heterogeneity by varying the value of $\alpha$ (see Sec. 4.3). Each participating client trains its local model for 4 local epochs using stochastic gradient descent (SGD) with a learning rate of 0.1. We adopt a partial client participation setting, where the server randomly selects $M = 10$ clients to participate in each communication round. The total number of communication rounds is set to 300.

**Attack Methods.** To evaluate the effectiveness of the proposed AUPF, we consider four representative backdoor attacks in federated learning: BadNet (Gu et al., 2017), DBA (Xie et al., 2020), Neurotoxin (Zhang et al., 2022b), and Layer-wise Poisoning (LP) (Zhuang et al., 2024). BadNet and DBA are *data-level* poisoning attacks (Huang et al., 2023b), where malicious clients implant triggers by modifying only the raw training samples. In contrast, *Neurotoxin and LP are algorithm-level attacks that additionally manipulate the training procedure (*e.g.*, identifying vulnerable neurons or perturbing specific layers), thereby producing more stealthy backdoors.*

Table 1: Defense performance (%) on different datasets under the non-IID setting ($\alpha = 0.5$). The best and runner-up results in unsupervised methods are highlighted with **bold** and underline, respectively.

| Attack / Defense | BadNet | | | DBA | | | Neurotoxin | | | LP | | |
|---|---|---|---|---|---|---|---|---|---|---|---|---|
| | ACC ↑ | ASR ↓ | RPG ↑ | ACC ↑ | ASR ↓ | RPG ↑ | ACC ↑ | ASR ↓ | RPG ↑ | ACC ↑ | ASR ↓ | RPG ↑ |
| **CIFAR-10** | | | | | | | | | | | | |
| FedAvg | 89.60 | 99.93 | N.A. | 89.36 | 98.27 | N.A. | 89.52 | 99.94 | N.A. | 89.71 | 79.44 | N.A. |
| FoolsGold | 89.53 | 99.96 | −0.10 | 89.12 | 99.65 | −1.62 | 89.04 | 99.95 | −0.48 | 89.49 | 91.39 | −12.17 |
| RLR | 84.72 | 99.94 | −4.89 | 84.32 | 97.13 | −3.91 | 84.61 | 99.92 | −4.88 | 84.69 | 63.70 | 10.72 |
| RFA | 71.16 | 30.78 | 50.71 | 70.06 | 32.29 | 46.69 | 70.14 | 28.72 | 51.85 | 77.23 | 92.08 | −25.12 |
| Multi-Krum | 86.93 | 99.92 | −2.65 | 86.86 | 97.00 | −1.22 | 86.74 | 99.87 | −2.71 | 87.87 | 92.87 | −15.28 |
| DeepSight | 85.66 | 99.93 | −3.93 | 85.73 | 98.40 | −3.76 | 88.11 | 99.94 | −1.40 | 88.45 | 92.20 | −14.02 |
| FLAME | 83.34 | 99.96 | −6.28 | 83.26 | 99.43 | −7.26 | 81.65 | 99.61 | −7.54 | 81.95 | 91.21 | −19.53 |
| Multi-metrics | 83.00 | 99.98 | −6.64 | 82.40 | 94.42 | −3.11 | 82.58 | 99.77 | −6.76 | 85.20 | 86.39 | −11.45 |
| Snowball | 87.01 | 99.98 | −2.64 | 86.96 | 99.09 | −3.22 | 87.32 | 99.97 | −2.23 | 87.27 | 95.83 | −18.83 |
| Snowball⊖ | 83.54 | 99.79 | −5.91 | 84.63 | 71.82 | 21.73 | 84.01 | 99.99 | −5.55 | 85.60 | 96.39 | −21.06 |
| FDCR | 86.28 | 47.55 | 49.07 | 86.52 | 49.29 | 46.14 | 86.67 | 28.01 | 69.09 | 88.13 | 44.22 | 33.64 |
| FLIP | 86.84 | 90.51 | 6.67 | 87.25 | 17.69 | 78.47 | 87.54 | 85.49 | 12.48 | 86.84 | 43.83 | 32.73 |
| Lockdown | 83.04 | 33.53 | 59.85 | 83.68 | 6.13 | 86.46 | 83.49 | 34.84 | 59.08 | 85.46 | 24.25 | 50.94 |
| AUPF | 84.61 | 4.96 | **89.99** | 85.66 | 2.03 | **92.55** | 84.49 | 4.80 | **90.12** | 84.97 | 8.00 | **66.70** |
| **FashionMNIST** | | | | | | | | | | | | |
| FedAvg | 90.95 | 98.87 | N.A. | 90.95 | 98.86 | N.A. | 90.76 | 98.77 | N.A. | 90.85 | 97.58 | N.A. |
| FoolsGold | 90.80 | 98.81 | −0.09 | 90.64 | 97.47 | 1.09 | 90.55 | 98.54 | 0.01 | 90.75 | 97.37 | 0.11 |
| RLR | 82.71 | 88.22 | 2.42 | 73.39 | 0.02 | 81.28 | 82.82 | 88.47 | 2.37 | 62.39 | 0.53 | 68.59 |
| RFA | 85.23 | 96.10 | −2.95 | 85.47 | 93.45 | −0.07 | 86.60 | 98.75 | −4.14 | 85.33 | 97.27 | −5.22 |
| Multi-Krum | 89.95 | 99.07 | −1.21 | 89.91 | 98.99 | −1.17 | 90.16 | 98.81 | −0.64 | 89.80 | 96.70 | −0.17 |
| DeepSight | 90.63 | 95.21 | 3.34 | 90.66 | 97.26 | 1.31 | 90.88 | 98.78 | 0.10 | 90.55 | 96.52 | 0.76 |
| FLAME | 88.27 | 99.34 | −3.14 | 87.77 | 77.77 | 17.91 | 87.71 | 99.18 | −3.46 | 88.02 | 96.52 | −1.78 |
| Multi-metrics | 87.78 | 98.91 | −3.21 | 89.48 | 81.81 | 15.58 | 88.58 | 99.14 | −2.55 | 88.60 | 95.46 | −0.13 |
| Snowball | 90.78 | 98.59 | 0.11 | 90.27 | 0.86 | **97.32** | 90.30 | 97.62 | 0.70 | 89.97 | 98.09 | −1.39 |
| Snowball⊖ | 89.47 | 95.46 | 1.93 | 89.85 | 0.81 | 96.96 | 89.31 | 95.65 | 1.67 | 88.45 | 98.14 | −2.97 |
| FDCR | 89.84 | 92.29 | 5.47 | 90.49 | 71.02 | 27.37 | 90.23 | 89.32 | 8.92 | 90.10 | 87.00 | 9.83 |
| FLIP | 86.30 | 64.12 | 30.10 | 73.38 | 10.59 | 75.71 | 84.62 | 49.13 | 43.50 | 82.50 | 40.16 | 49.07 |
| Lockdown | 89.12 | 43.59 | 53.45 | 88.94 | 58.28 | 38.56 | 88.29 | 27.84 | 68.46 | 89.73 | 91.83 | 4.63 |
| AUPF | 88.63 | 8.26 | **88.30** | 88.19 | 26.41 | 69.69 | 88.06 | 11.09 | **84.98** | 88.46 | 20.94 | **74.25** |
| **CIFAR-100** | | | | | | | | | | | | |
| FedAvg | 66.89 | 99.60 | N.A. | 66.54 | 99.24 | N.A. | 66.87 | 99.60 | N.A. | 67.57 | 90.81 | N.A. |
| FoolsGold | 66.53 | 99.54 | −0.30 | 67.06 | 98.97 | 0.79 | 66.54 | 99.71 | −0.45 | 67.00 | 93.39 | −3.15 |
| RLR | 61.23 | 98.78 | −4.84 | 61.24 | 97.39 | −3.45 | 61.75 | 99.08 | −4.60 | 61.33 | 41.05 | 43.53 |
| RFA | 42.27 | 0.69 | 74.29 | 42.32 | 0.78 | 74.24 | 43.03 | 0.99 | 74.76 | 44.36 | 86.87 | −19.26 |
| Multi-Krum | 58.71 | 42.38 | 49.04 | 58.84 | 46.64 | 44.90 | 58.68 | 64.75 | 26.65 | 60.51 | 86.74 | −2.99 |
| DeepSight | 65.27 | 99.81 | −1.83 | 66.20 | 99.58 | −0.69 | 65.03 | 99.73 | −1.97 | 66.73 | 90.78 | −0.80 |
| FLAME | 58.75 | 99.55 | −8.09 | 56.64 | 88.63 | 0.71 | 59.97 | 99.75 | −7.06 | 56.71 | 85.38 | −5.43 |
| Multi-metrics | 62.29 | 99.75 | −4.75 | 62.65 | 99.24 | −3.89 | 62.30 | 99.64 | −4.62 | 62.30 | 86.17 | −0.62 |
| Snowball | 64.57 | 98.35 | −1.08 | 63.40 | 94.34 | 1.76 | 63.62 | 95.91 | 0.43 | 65.08 | 86.68 | 1.64 |
| Snowball⊖ | 60.30 | 0.38 | **92.63** | 60.62 | 0.62 | **92.70** | 60.20 | 0.67 | **92.26** | 61.30 | 82.60 | 1.94 |
| FDCR | 64.92 | 94.53 | 3.10 | 64.82 | 78.05 | 19.46 | 64.77 | 93.93 | 3.57 | 64.96 | 44.71 | 43.49 |
| Lockdown | 57.80 | 15.50 | 75.02 | 61.31 | 38.62 | 55.39 | 57.87 | 8.68 | 81.91 | 55.40 | 32.30 | 46.34 |
| AUPF | 58.75 | 8.27 | 83.19 | 58.40 | 5.86 | 85.24 | 58.54 | 16.39 | 74.88 | 58.26 | 8.53 | **72.97** |

For evaluating AUPF, we assume an adaptive attacker that tailors local training to our defense, with malicious clients performing updates as defined in Eq. (S1). For baseline defenses, we use the standard (non-adaptive) versions of each attack. Further implementation and training details are provided in Appendix B.2.

**Defense Methods.** We use the vanilla FedAvg algorithm (McMahan et al., 2017) as a baseline, where no specific defense strategy is applied. We also compared the proposed method with 11 state-of-the-art or representative backdoor defense methods in federated learning: FoolsGold (Fung et al., 2020), RLR (Özdayi et al., 2021), RFA (Pillutla et al., 2022), Multi-Krum (Blanchard et al., 2017), DeepSight (Rieger et al., 2022), FLAME (Nguyen et al., 2022), Multi-metrics (Huang et al., 2023a), Snowball (Qin et al., 2024), FDCR (Huang et al., 2024), Lockdown (Huang et al., 2023b), and FLIP (Zhang et al., 2023b). Following the original Snowball paper (Qin et al., 2024), we also consider a variant, denoted Snowball⊖, which retains only the bottom-up election mechanism from the full Snowball framework. By selecting fewer clients during aggregation, Snowball⊖ typically achieves stronger backdoor defense compared to the full Snowball method. Additional details of these defense methods are provided in Appendix B.3.

**Evaluation Metrics.** To assess the effectiveness of federated learning backdoor defenses, we employ three metrics: 1) **Main Task Accuracy (ACC)**: the accuracy on clean test inputs (*i.e.*, without triggers); 2) **Attack Success Rate (ASR)**: the accuracy on backdoor test inputs (*i.e.*, containing the trigger and labeled with the target class); 3) **Real Performance Gain (RPG)**: a composite metric that captures both defense effectiveness and cost by jointly considering ACC and ASR. Formally, we define RPG := $\Delta$ASR $- \Delta$ACC where $\Delta$ASR denotes the reduction in ASR after applying the defense (compared to FedAvg), and $\Delta$ACC denotes the corresponding reduction in ACC. A larger RPG thus indicates a more favorable trade-off between mitigating backdoor attacks and preserving main-task performance. All metrics are evaluated based on the average results from the last 10 communication rounds.

**Attack Setup.** To simulate backdoor attacks, we assume the attacker compromises 20% of the total $N$ clients. This attacker proportion is preserved in every communication round (*i.e.*, 20% of the $M$ participating clients are malicious in each round). Following Özdayi et al. (2021), each malicious client randomly poisons 20% of its local data. We evaluate robustness under varying levels of data heterogeneity and poisoning ratios in Sec. 4.3. Consistent with Lockdown (Özdayi et al., 2021; Huang et al., 2023b), we use a "plus" pattern as the backdoor trigger. For all three datasets the target label is fixed to "7", corresponding to "Horse" in CIFAR-10, "Aquarium Fish" in CIFAR-100, and "Sneaker" in FashionMNIST.

## 4.2 EFFECTIVENESS

Table 1 reports the defense performance of AUPF compared with 11 state-of-the-art approaches on CIFAR-10, FashionMNIST, and CIFAR-100. Since the computational complexity of FLIP scales quadratically with the number of classes, it becomes impractical for datasets with many classes; hence, we report its results only on CIFAR-10 and FashionMNIST.

Across all three datasets and four attacks, AUPF consistently achieves the strongest overall robustness, combining *low ASR* with *competitive ACC*, which yields the *highest RPG* in almost all cases. On CIFAR-10, AUPF reduces ASR to 4.96/2.03/4.80/8.00% for BadNet/DBA/Neurotoxin/LP while maintaining ACC around 84–86%, resulting in RPG values close to 90 for the first three attacks and 66.7 for LP. Similar trends are observed on FashionMNIST (ASR 8.26/26.41/11.09/20.94%, ACC ≈88%) and CIFAR-100 (ASR 8.27/5.86/16.39/8.53%, ACC ≈58–59%). We further note that *algorithm-level attacks* (Neurotoxin, LP) are significantly more challenging than *data-level* ones (BadNet, DBA): while several baselines suffer ASR near or above 90% under LP on CIFAR-10, AUPF limits ASR to just 8.00%. Defenses such as Lockdown and FLIP can be effective in specific scenarios (e.g., DBA), but their performance often degrades or becomes unstable across other attacks or datasets, resulting in unfavorable RPG. Overall, AUPF achieves a superior balance between robustness and utility in the *realistic* non-IID regime, underscoring its practical deployability.

On CIFAR-100, AUPF shows a small ACC drop relative to FedAvg, but this is a common robustness–utility trade-off on large-class datasets with complex decision boundaries. Defenses that substantially reduce ASR (e.g., RFA, FLAME, Multi-Krum) exhibit similar or larger ACC reductions. Our hyperparameter study (Sec. 4.4) further shows that AUPF is stable across $\lambda_2$, indicating that the ACC gap is not due to sensitivity. Notably, AUPF achieves the best or near-best RPG across all four attacks, highlighting its strong overall robustness under this challenging setting.

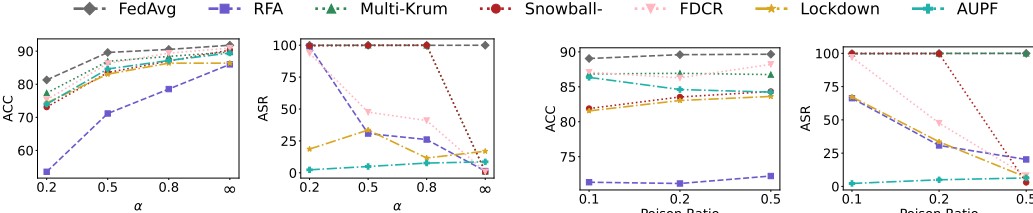

Figure 2: Sensitivity of different methods to data heterogeneity on CIFAR-10, where $\alpha = \infty$ corresponds to IID clients and smaller $\alpha$ values indicate greater heterogeneity.

Figure 3: Sensitivity of different methods to poison ratios on CIFAR-10 under the non-IID setting ($\alpha = 0.5$).

### 4.3 SENSITIVITY

We evaluate the sensitivity of the proposed method to data heterogeneity, poison ratio, attacker ratio, and the number of clients. Experiments are conducted on CIFAR-10 using BadNet as the attack method to illustrate the results.

**Sensitivity to Data Heterogeneity.** We evaluate our method under varying levels of data heterogeneity by adjusting the Dirichlet parameter $\alpha$, smaller $\alpha$ values correspond to higher degrees of heterogeneity. As shown in Fig. 2, our AUPF consistently achieves the lowest ASR in non-IID settings, even under highly heterogeneous conditions ($\alpha = 0.2$). In contrast, most existing defenses perform reasonably well in the IID setting but deteriorate rapidly as heterogeneity increases, and nearly collapse when $\alpha = 0.2$, with ASR approaching 100%. Since real-world federated learning systems are inherently non-IID, these results highlight the practical robustness of our method in realistic deployment scenarios, making it substantially more applicable than prior defenses.

**Sensitivity to Poison Ratio.** We evaluate the sensitivity of AUPF to poisoning ratios $\{0.1, 0.2, 0.5\}$ on CIFAR-10 under a non-IID setting ($\alpha = 0.5$), with results shown in Fig. 3. The ASR of FedAvg remains close to 100% across all ratios, indicating complete vulnerability. As observed, AUPF consistently maintains low ASR across poison ratios, whereas existing defenses degrade markedly as the ratio decreases. Moreover, defenses that rely on distinguishing benign from malicious updates perform relatively better at higher ratios, where the differences between the two types of updates are more pronounced.

In Appendix Sec. A.3, we further analyze the performance of robust aggregation–based defenses under different combinations of data heterogeneity and poisoning ratios, aiming to provide diagnostic insight into why these defenses succeed in some scenarios but fail in others. We also present additional evaluations, including sensitivity to attacker ratio, number of clients, and trigger type, in Appendix Sec. A.

### 4.4 HYPER-PARAMETER SENSITIVITY ANALYSIS AND ABLATION STUDY

In this section, we perform hyper-parameter sensitivity analysis for AUPF. The study is conducted on the CIFAR-10 under the default experiment setting unless otherwise specified.

**Perturbation Strength $\rho$.** We fix the attacker's trigger and vary the defender's perturbation bound as $\rho$. Fig. 4 (a) shows that larger $\rho$ values substantially reduce ASR while maintaining relatively stable ACC. This indicates that increasing $\rho$ enhances the defense effectiveness with only a moderate impact on the main task performance. *Notably, AUPF degenerates to FedAvg when $\rho = 0$. Compared with this baseline, the results clearly demonstrate the effectiveness of the proposed AUPF in defending against backdoor attacks.*

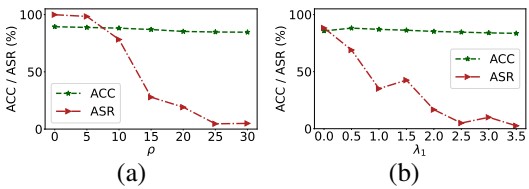

(a)  (b)

Figure 4: Impact of the hyper-parameters: (a) $\rho$ and (b) $\lambda_1$ on AUPF performance.

**Adversarial Unlearning $\lambda_1$.** We tune the adversarial unlearning weight $\lambda_1$, which controls the strength of the unlearning objective. As shown in Fig. 4 (b), once $\lambda_1$ exceeds a small threshold, AUPF effectively suppresses the backdoor while keeping ACC within a reasonable range. Increasing $\lambda_1$ further introduces a trade-off between ASR and ACC: ASR continues to decrease, whereas ACC exhibits a slight decline. Hence, a moderate $\lambda_1$ achieves strong backdoor mitigation with minimal impact on the main task.

Table 2: Impact of the hyper-parameter $\lambda_2$ on AUPF performance.

| $\lambda_2$ | 0 | 0.5 | 0.8 | 1.0 | 1.2 | 1.5 |
|---|---|---|---|---|---|---|
| ACC | 86.17 | 85.15 | 85.07 | 84.61 | 84.21 | 84.73 |
| ASR | 18.95 | 21.46 | 16.21 | 4.96 | 6.30 | 9.71 |

**Feature Unlearning $\lambda_2$.** Table 2 evaluates the effect of the feature-unlearning weight $\lambda_2$ on AUPF. When $\lambda_2 < 1$ is too small, ASR remains high (around 20%). Increasing $\lambda_2$ substantially suppresses the backdoor, while ACC decreases only slightly. These results demonstrate the effectiveness of feature unlearning and support choosing a moderate $\lambda_2$.

## 4.5 TRAINING TIME COMPARISON

Table 3 shows that Lockdown adds only slight overhead compared with FedAvg, whereas FLIP is computationally prohibitive, requiring more than two orders of magnitude longer. AUPF moderately increases training time due to adversarial unlearning, yet its overhead remains far smaller than that of FLIP, offering a practical balance between robustness and efficiency.

## 4.6 ADDITIONAL EXPERIMENTS AND RESULTS

We further evaluate AUPF under an adaptive unlearning-targeted attack A3FL (Zhang et al., 2023a) and a persistence-oriented attack Chameleon (Dai & Li, 2023); see Appendix A.8. We also compare AUPF with the recently proposed server-side robust aggregation-based defense AlignIns (Xu et al., 2025) in Appendix A.9. In addition, we evaluate the scalability of AUPF to larger models (ResNet-50) and larger client populations (150 and 200 clients) in Appendix A.10 and Appendix A.11, respectively. Taken together, these additional experiments provide a more comprehensive evaluation of AUPF.

## 5 CONCLUSION

In this work, we studied backdoor defense in federated learning under a practical yet challenging setting, where malicious clients may launch continuous and adaptive attacks in highly non-IID environments, and the server has no access to auxiliary data. To address these challenges, we proposed a defense framework based on adversarial unlearning of poisoned features (AUPF). Extensive experiments across diverse datasets and attack scenarios demonstrate that AUPF consistently achieves strong robustness and surpasses existing defenses under these conditions.

Table 3: Average per-round training time (s) of benign clients over 100 communication rounds with 4 local epochs on an NVIDIA RTX3090 (24GB) GPU.

| Dataset | FedAvg | FLIP | Lockdown | AUPF |
|---|---|---|---|---|
| CIFAR-10 | 3.15 | 307.04 | 3.64 | 7.83 |
| FashionMNIST | 1.07 | 164.49 | 1.18 | 2.78 |

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

# A  ADDITIONAL EXPERIMENTS AND RESULTS

Table S1: Results of adaptive attacks against FDCR (Huang et al., 2024) and Lockdown (Huang et al., 2023b) on the CIFAR-10 dataset.

| | | IID | | | | Non-IID ($\alpha = 0.5$) | | | |
| | | FDCR | | Lockdown | | FDCR | | Lockdown | |
| Attack | Defense | ACC ↑ | ASR ↓ | ACC ↑ | ASR ↓ | ACC ↑ | ASR ↓ | ACC ↑ | ASR ↓ |
|---|---|---|---|---|---|---|---|---|---|
| BadNet | Normal | 91.42 | 0.89 | 90.41 | 9.70 | 88.21 | 8.12 | 85.59 | 18.96 |
| | Adaptive | 91.42 | 99.85 | 90.09 | 99.99 | 85.25 | 99.92 | 85.01 | 99.99 |
| DBA | Normal | 91.06 | 0.88 | 90.46 | 6.92 | 87.80 | 1.99 | 86.07 | 6.15 |
| | Adaptive | 91.04 | 92.71 | 90.23 | 99.95 | 87.35 | 94.64 | 85.49 | 98.88 |
| Neurotoxin | Normal | 91.29 | 0.86 | 90.42 | 11.42 | 88.66 | 2.60 | 85.75 | 21.96 |
| | Adaptive | 91.17 | 99.54 | 90.44 | 100.00 | 86.82 | 99.93 | 84.95 | 99.98 |
| LP | Normal | 91.16 | 0.82 | 89.20 | 48.33 | 86.81 | 3.70 | 85.84 | 35.67 |
| | Adaptive | 91.26 | 36.55 | 89.17 | 76.18 | 87.49 | 10.12 | 85.16 | 49.91 |

## A.1  ADAPTIVE ATTACKS AGAINST FDCR (HUANG ET AL., 2024) AND LOCKDOWN (HUANG ET AL., 2023B)

As discussed in the Sec. 1 of main text, defense mechanisms that rely on *client cooperation* are inherently vulnerable to adaptive attacks, as malicious clients can deliberately bypass or manipulate the prescribed strategies. FDCR (Huang et al., 2024) and Lockdown (Huang et al., 2023b) are representative examples of such approaches. In this section, we present experimental results showing that carefully crafted adaptive attacks can successfully circumvent both defenses. The results are summarized in Table S1, with details of the adaptive attacks provided below. Importantly, for each attack type, the *normal* and *adaptive* variants are evaluated under identical hyperparameter settings, ensuring a fair comparison.

- **Adaptive Attack on FDCR (Huang et al., 2024)**: FDCR requires each client to compute and submit a Fisher Information Matrix (FIM), which quantifies the importance of each parameter in the local model. During aggregation, the server uses the FIM to reweight client gradient updates and identifies clients with significant gradient divergence as potential adversaries, subsequently exclude their contributions. However, the original paper makes the unrealistic assumption that malicious clients also faithfully compute the FIM using their local models and data. To exploit this, we design an adaptive attack tailored to FDCR. *The attacker computes the FIM using the global model downloaded from the server and clean local data*. This causes the submitted FIM to appear benign, allowing the malicious client to evade detection. As a result, the server fails to detect the backdoor updates, enabling the attack to succeed.
- **Adaptive Attack on Lockdown (Huang et al., 2023b)**: In Lockdown, each client is restricted to perform local training within an isolated subspace that captures parameters important to its local data. Clients must also upload this subspace to the server for consensus-based voting. However, the original paper similarly assumes that even malicious clients honestly identify their subspaces using local data, which is an unrealistic premise in adversarial settings. To challenge this, we design an adaptive attack specifically targeting Lockdown. *The attacker first leverages clean local data to recover a meaningful subspace, then randomly prunes the mask to satisfy the server's sparsity requirements, and finally projects its gradient updates into this subspace*. This strategy enables backdoor gradients to evade pruning by the consensus mechanism, effectively preserving the malicious updates.

## A.2  EFFECTIVENESS OF FEDERATED LEARNING DEFENSES UNDER IID SETTINGS

To assess the impact of data heterogeneity on federated learning defenses, we conduct experiments on the CIFAR-10 dataset under an IID client distribution, keeping all other experimental configurations consistent with Table 1. The results in Table S2 show that, compared with the non-IID setting, most defenses (e.g., FoolsGold, RFA, Multi-Krum, and DeepSight) achieve strong robustness under

Table S2: Comparison of defense performance (%) on CIFAR-10 under the IID setting between the proposed method and 11 state-of-the-art defenses. All other experimental configurations are consistent with those in Table 1.

| Attack / Defense | BadNet ACC ↑ | ASR ↓ | RPG ↑ | DBA ACC ↑ | ASR ↓ | RPG ↑ | Neurotoxin ACC ↑ | ASR ↓ | RPG ↑ | LP ACC ↑ | ASR ↓ | RPG ↑ |
|---|---|---|---|---|---|---|---|---|---|---|---|---|
| FedAvg | 91.79 | 99.95 | N.A. | 91.91 | 99.85 | N.A. | 91.78 | 99.95 | N.A. | 92.22 | 93.86 | N.A. |
| FoolsGold | 91.58 | 0.81 | 98.92 | 91.42 | 0.76 | 98.60 | 91.59 | 0.70 | 99.05 | 91.72 | 0.90 | 92.47 |
| RLR | 90.33 | 99.78 | −1.30 | 90.22 | 97.77 | 0.39 | 90.26 | 99.81 | −1.39 | 90.54 | 59.31 | 32.88 |
| RFA | 86.00 | 1.39 | 92.77 | 84.79 | 1.83 | 90.91 | 86.48 | 1.17 | 93.48 | 87.99 | 66.51 | 23.13 |
| Multi-Krum | 89.73 | 1.08 | 96.80 | 89.83 | 0.77 | 97.01 | 89.73 | 1.08 | 96.81 | 90.45 | 95.96 | −3.87 |
| DeepSight | 89.96 | 1.55 | 96.57 | 90.13 | 1.14 | 96.94 | 90.23 | 2.58 | 95.81 | 92.10 | 83.48 | 10.26 |
| FLAME | 88.50 | 99.89 | −3.24 | 90.64 | 97.73 | 0.85 | 90.24 | 99.95 | −1.55 | 90.64 | 75.32 | 16.96 |
| Multi-metrics | 91.19 | 99.87 | −0.53 | 90.88 | 99.92 | −1.10 | 90.86 | 99.92 | −0.89 | 90.93 | 90.35 | 2.23 |
| Snowball | 90.91 | 98.47 | 0.59 | 91.07 | 74.17 | 24.85 | 91.37 | 99.36 | 0.17 | 91.33 | 65.17 | 27.81 |
| Snowball⊖ | 90.31 | 0.83 | 97.64 | 90.38 | 0.65 | 97.68 | 90.42 | 0.74 | 97.85 | 90.09 | 0.68 | 91.06 |
| FDCR | 90.92 | 0.84 | 98.23 | 91.36 | 0.81 | 98.49 | 90.96 | 1.05 | 98.08 | 91.62 | 23.48 | 69.78 |
| FLIP | 90.78 | 99.25 | −0.32 | 90.58 | 83.15 | 15.38 | 90.98 | 97.91 | 1.23 | 91.03 | 37.14 | 55.54 |
| Lockdown | 86.37 | 16.95 | 77.57 | 90.04 | 14.96 | 83.03 | 86.58 | 13.24 | 81.51 | 89.17 | 17.85 | 72.97 |
| AUPF | 89.44 | 8.62 | 88.98 | 89.22 | 1.05 | 96.12 | 89.08 | 4.35 | 92.90 | 89.15 | 13.18 | 77.62 |

IID conditions. This indicates that their effectiveness is highly sensitive to the degree of client data heterogeneity, whereas AUPF maintains consistent and reliable robustness across both IID and non-IID distributions.

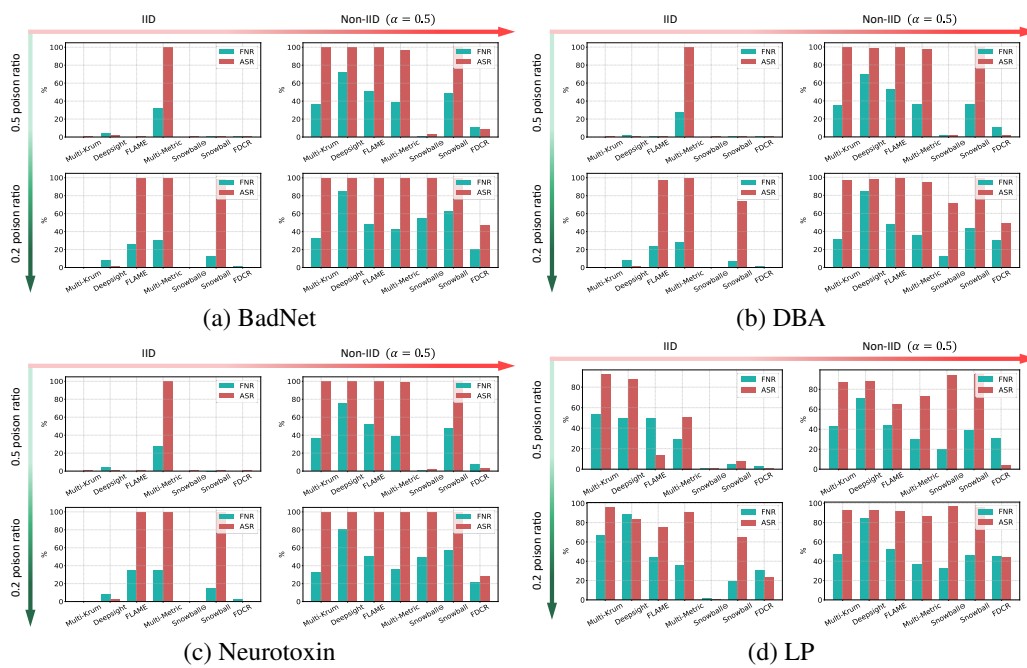

Figure S1: The FNR (false negative rate) and ASR of robust aggregation-based defenses on the CIFAR-10 dataset under the four attack strategies.

## A.3  ANALYZING THE PERFORMANCE OF ROBUST AGGREGATION-BASED DEFENSES

As shown in Table 1, robust aggregation-based defenses generally perform poorly across various datasets and attack methods. These defenses are designed to identify and exclude malicious clients from aggregation, thereby mitigating the backdoor effect in the global model. However, our analysis reveals that their effectiveness is highly sensitive to the degree of data heterogeneity across clients

and the poison ratio of malicious clients. Specifically, when the data distribution is highly non-IID or the poison ratio is low, malicious clients may evade detection and be mistakenly included in the aggregation process. Consequently, backdoor patterns are embedded in the global model, leading to a high attack success rate (ASR).

To validate this observation, we conduct experiments on the CIFAR-10 dataset by varying either the data heterogeneity (IID or non-IID with $\alpha = 0.5$) or the poison ratio (0.2 or 0.5), while keeping all other settings consistent with the default setting. Since the goal of the defense is to identify and exclude malicious clients from aggregation, we define malicious clients as positive and benign clients as negative. Accordingly, the false negative rate (FNR) serves as a key metric to evaluate detection performance. *A higher FNR indicates that more malicious clients are misclassified as benign and retained in the aggregation, thereby allowing more backdoor influence to be injected into the global model.* Fig. S1 presents the FNR and ASR of the defenses under the BadNet, DBA, Neurotoxin, and LP attacks, respectively. Note that the FNR is averaged over all 300 communication rounds. We have the follow observation:

- *The ASR of these defenses is generally proportional to the FNR across different attack methods, levels of data heterogeneity, and poisoning ratios on malicious clients.* As observed in the bottom-right corners of Fig. S1 (a), (b), (c), and (d), these defenses exhibit a very high FNR. This directly explains their poor performance in Table 1 (*i.e.*, 0.2 poison ratio and Non-IID with $\alpha = 0.5$), as the backdoor model is retained in the global model due to the inclusion of undetected malicious clients.
- *The FNR of these methods increases with greater data heterogeneity or lower poison ratios on malicious clients.* This is because in highly non-IID settings or under low poison ratios, the differences between benign and malicious updates become increasingly subtle, making it harder for the server to reliably identify adversarial clients.
- *Most of these defenses fail to mitigate the LP attack even under a 0.5 poisoning ratio and IID conditions.* This is likely because LP selectively poisons only the backdoor-critical layers and dynamically adjusts the selection or scale of targeted layers to evade aggregation filters, thereby achieving highly effective yet stealthy backdoors.

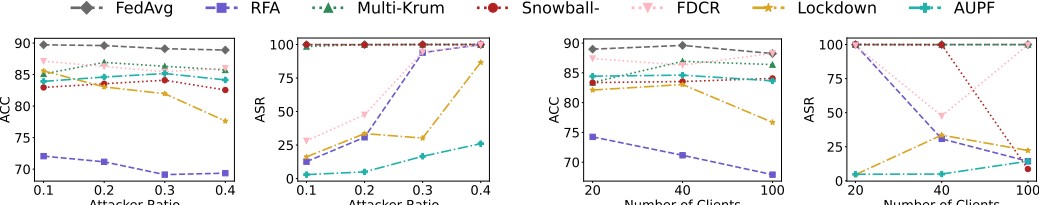

Figure S2: Sensitivity of different methods to the attacker ratio on CIFAR-10 under the non-IID setting ($\alpha = 0.5$).

Figure S3: Sensitivity of different methods to the number of clients on CIFAR-10 under the non-IID setting ($\alpha = 0.5$).

### A.4 SENSITIVITY TO THE ATTACKER RATIO AND THE NUMBER OF CLIENTS

**Sensitivity to the Attacker Ratio.** We evaluate the sensitivity of AUPF to the attacker ratio by varying it across $\{0.1, 0.2, 0.3, 0.4\}$ under the non-IID setting ($\alpha = 0.5$). The results are presented in Fig. S2. Overall, all defenses exhibit noticeable performance degradation as the attacker ratio increases. This trend is understandable, since a higher attacker ratio effectively corresponds to a larger attack budget. Nevertheless, following key observations can be made: 1) AUPF exhibits the lowest sensitivity to changes in the attacker ratio; 2) AUPF consistently achieves the lowest ASR across all attacker ratios; and 3) As the attacker ratio increases, the performance degradation remains **gradual rather than abrupt**, indicating that AUPF does not rely on a strong benign-majority assumption.

**Sensitivity to the Number of Clients.** Fig. S3 shows the performance of the proposed AUPF on CIFAR-10 under the default setting with varying numbers of clients: 20, 40, and 100. As observed, our AUPF consistently achieves the lowest ASR while keeping the ACC degradation within a reasonable range. In contrast, many defenses (*e.g.*, RFA, Snowball⊖, FDCR, Lockdown) exhibit high sensitivity to the total number of clients and fail to defend effectively when the number of clients is either too small or too large.

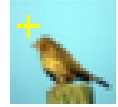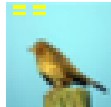

Figure S4: Examples of trigger patterns. From left to right: a clean input, an input poisoned with the "plus" pattern, and an input poisoned with the "square" pattern.

Table S3: Evaluation of defense methods against different backdoor triggers on the CIFAR-10 dataset.

| Trigger
Defense | Plus Pattern | | | Square Pattern | | |
|---|---|---|---|---|---|---|
| | ACC ↑ | ASR ↓ | RPG ↑ | ACC ↑ | ASR ↓ | RPG ↑ |
| FedAvg | 89.60 | 99.93 | N.A. | 89.21 | 99.94 | N.A. |
| RFA | 71.16 | 30.78 | 50.71 | 70.12 | 45.32 | 35.54 |
| Multi-Krum | 86.93 | 99.92 | −2.65 | 86.45 | 99.88 | −2.69 |
| Snowball1 | 83.54 | 99.79 | −5.91 | 83.58 | 99.97 | −5.65 |
| FDCR | 86.28 | 47.55 | 49.07 | 86.79 | 79.50 | 18.03 |
| Lockdown | 83.04 | 33.53 | 59.85 | 80.83 | 8.72 | 82.84 |
| AUPF | 84.61 | 4.96 | 89.99 | 84.11 | 12.53 | 82.31 |

## A.5 SENSITIVITY ANALYSIS ON TRIGGER PATTERNS

In this section, we conduct experiments to analyze the sensitivity of the proposed method to different trigger patterns. Fig. S4 illustrates the two trigger patterns used in our experiments. Specifically, the "plus" pattern is adopted for all experiments presented in the main text. As shown in Table S3, our method consistently achieves the best performance in term of RPG for both trigger patterns, whereas RFA, FDCR, and Lockdown exhibit notable performance variations across different triggers. These results demonstrate the robustness of our method to variations in trigger patterns.

## A.6 EFFECTIVENESS OF POISONED FEATURE UNLEARNING

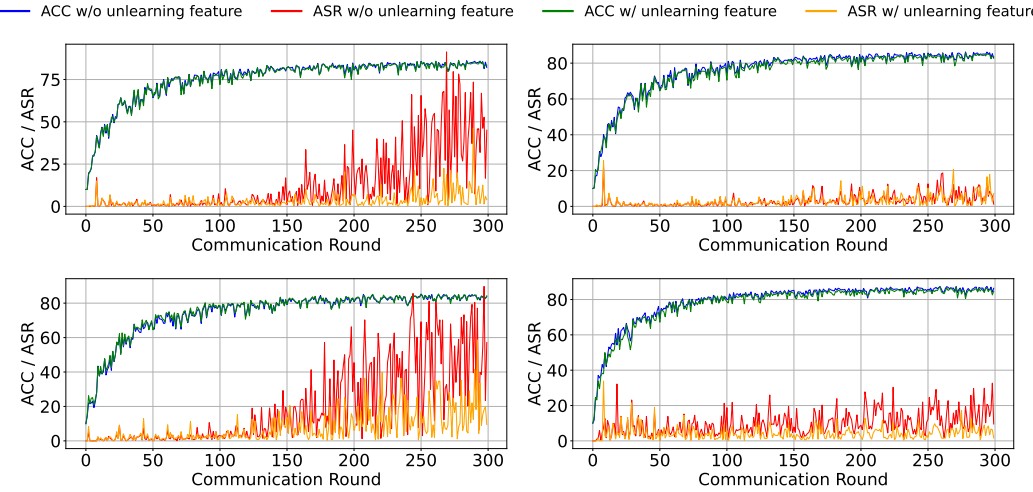

Figure S5: Effectiveness of poisoned feature unlearning on CIFAR-10. From left to right, the attack methods shown are BadNet, DBA, Neurotoxin, and LP. Both variants achieve similar ACC, but without poisoned feature unlearning the ASR fluctuates sharply across communication rounds, whereas incorporating it stabilizes the ASR and enhances robustness.

In Sec. 3, we highlighted that the dynamic evolution of the federated global model poses a key challenge for defense: although adversarial training enables benign clients to locally suppress backdoors, malicious clients can continually reintroduce them into the global model through repeated

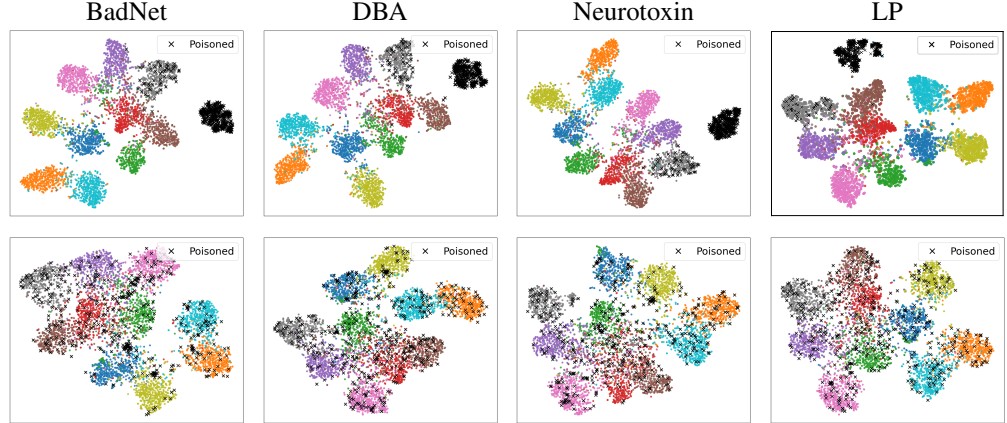

Figure S6: t-SNE visualizations of sample representations on the CIFAR-10 dataset under four different backdoor attack methods (BadNet, DBA, Neurotoxin, and LP). The top row shows results without defense (FedAvg), where poisoned samples collapse into a distinct cluster. The bottom row shows results with the proposed defense (AUPF), where poisoned samples realign with their original clusters, indicating that the poisoned features have been effectively unlearned.

participation and aggregation. To address this, we propose adversarial unlearning of poisoned features.

We now empirically evaluate its effectiveness by ablating the feature unlearning term in Eq. (6). Specifically, we compare the evolution of accuracy (ACC) and attack success rate (ASR) across communication rounds on CIFAR-10 under the default experimental setting, except with a 0.5 poison ratio. As shown in Fig. S5, both variants achieve comparable ACC, but their ASR trajectories differ substantially. Without poisoned feature unlearning, the ASR exhibits sharp fluctuations over rounds, reflecting instability in the global model under continual malicious influence. In contrast, incorporating poisoned feature unlearning yields consistently lower and more stable ASR, demonstrating enhanced robustness against persistent backdoor reinjection.

To further highlight the effectiveness of poisoned feature unlearning, we compare the t-SNE representations with and without defense. Fig. S6 presents the visualizations under four different attacks on CIFAR-10. Without defense (FedAvg), poisoned samples collapse into a distinct cluster. With the proposed defense (AUPF), these poisoned samples realign with their original clusters, indicating that the poisoned features have been effectively unlearned and the backdoor influence mitigated.

Table S4: Statistics of trigger perturbations and performance (%) of AUPF with the default parameter setting ($\rho = 30$) on CIFAR-10. The $\ell_2$-distance is computed as the mean difference between clean and poisoned image pairs across 10,000 test samples from the CIFAR-10 dataset.

| Trigger size ($n$) | #Pixels changed | $\ell_2$-distance (poisoned vs. clean) | ACC | ASR |
|---|---|---|---|---|
| 2 | 5 | 10.55 | 84.70 | 4.18 |
| 4 | 9 | 14.13 | 84.19 | 2.61 |
| 6 | 13 | 16.95 | 84.61 | 4.96 |
| 8 | 17 | 19.37 | 84.32 | 9.93 |
| 10 | 21 | 21.50 | 84.77 | 12.05 |
| 12 | 25 | 23.31 | 84.68 | 12.86 |
| 14 | 29 | 25.11 | 84.07 | 20.71 |
| 16 | 33 | 26.87 | 85.41 | 14.56 |
| 18 | 37 | 28.55 | 84.94 | 16.57 |

A.7 SENSITIVITY ANALYSIS ON PERTURBATION STRENGTH $\rho$

In Sec. 4.4, we demonstrated that larger values of $\rho$ substantially reduce ASR while maintaining stable ACC. Here, we further investigate the role of $\rho$ by fixing $\rho = 30$ for the defense and varying the attacker's trigger strength. Specifically, we adjust the arm length $n$ of the yellow "plus" trigger (see Fig. S4), where $n \in \{2, 4, 6, 8, 10, 12, 14, 16, 18\}$. Following prior works (Huang et al., 2023b; Özdayi et al., 2021), note that all main-text experiments adopt $n = 6$.

As reported in Table S4, ACC remains stable (84–85%), while ASR is consistently low for small triggers and increases only moderately with larger ones, peaking at 20.71% when $n = 14$. In most cases, ASR stays below 12%, indicating that even with a fixed defense budget, AUPF effectively mitigates a wide range of trigger sizes. Considering that CIFAR-10 images are only $32 \times 32$ pixels, triggers with $n \geq 14$ are already disproportionately large; in realistic threat scenarios, such oversized and visually conspicuous triggers are unlikely to be used due to stealthiness concerns. This highlights the practical robustness of our defense under real-world attack models.

A.8 ADDITIONAL RESULTS ON ADAPTIVE AND PERSISTENCE-ORIENTED ATTACKS (A3FL (ZHANG ET AL., 2023A) AND CHAMELEON (DAI & LI, 2023))

In this section, we further evaluate two additional advanced attacks: the adaptive unlearning-targeted A3FL (Zhang et al., 2023a) and the persistence-oriented Chameleon attack (Dai & Li, 2023).

A.8.1 ADDITIONAL RESULTS ON A3FL (ZHANG ET AL., 2023A)

Table S5: ACC/ASR (%) of different defenses on CIFAR-10 under the A3FL attack.

| FedAvg | FDCR | Snowball⊖ | Lockdown (*increased budget*) | AUPF | AUPF (*no adaptive attack*) |
|---|---|---|---|---|---|
| 88.63/99.99 | 86.85/99.77 | 82.33/100.00 | 71.40/77.14 | **80.34/36.18** | **84.39/22.40** |

**A3FL under continuous FL poisoning.** A3FL is designed for a late-phase poisoning scenario in which the attacker remains inactive until the main task has largely converged. To evaluate A3FL **fairly and compatibly** under our **continuous-poisoning threat model**, we adopt the **minimal modification needed for applicability**. All clients behave benignly for the first 100 rounds. At round 100, a single malicious client optimizes the trigger using A3FL and shares the resulting trigger with all other malicious clients. Once the trigger is fixed, all malicious clients initiate continuous poisoning for the remaining 200 rounds (as described in Sec. 4.1). In contrast, the original A3FL assumes that attacks are launched only during the last 100 rounds with random frequency. Therefore, this results in a substantially stronger attack compared to the original A3FL formulation. For AUPF, we also examine a variant in which the malicious clients perform standard poisoning training, without the adaptive attack described in Appendix B.1.

**Baseline defenses fail under the strengthened A3FL attack.** Under this strengthened A3FL attack, existing defenses significantly degrade. The results in terms of ACC/ASR (%) on CIFAR-10 are summarized in the Table S5. For Lockdown, the default configuration **fails completely (ASR ≈ 100%)**. We therefore increase its defense budget (anneal_ratio from 0.0001 to 0.1), but even with this strengthened configuration, the ASR only drops to **77%** and the ACC decreases to **71%**. Other defenses (FDCR, Snowball⊖) also completely fail, all producing **ASR above 99%**. In contrast, **AUPF remains highly robust under A3FL.**

These results show that 1) **AUPF substantially outperforms all existing defenses under A3FL**; 2) **AUPF remains robust even against an unlearning-targeted adaptive attack**; and 3) the non-adaptive attacker variant provides the **upper bound of AUPF** when the attacker does not perform adversarial training.

A.8.2 ADDITIONAL RESULTS ON CHAMELEON (DAI & LI, 2023)

**Chameleon** utilizes contrastive learning to amplify the backdoor effect, thereby enhancing its **long-term persistence**. We evaluate Chameleon to gain valuable insights into AUPF's robustness against stealthy, persistence-oriented attacks. In this experiment, we set the malicious client ratio to 10%

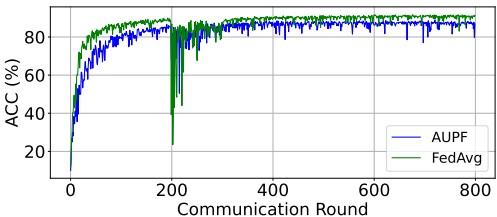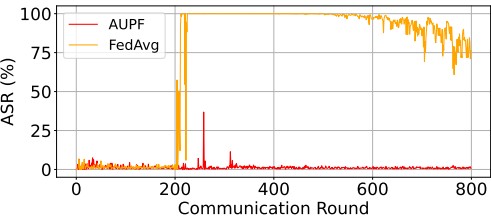

Figure S7: The ACC and ASR of AUPF on the CIFAR-10 dataset under the Chameleon attack.

(compared to 20% in our paper), as a 20% ratio would lead to excessively low ACC in this setting. **All other experimental settings follow Sec. 4.1 of our paper.**

To ensure comparability, we follow the **original Chameleon attack schedule**: the adversary poisons during **rounds 200 to 300** and remains inactive during **rounds 300 to 800**. This setup is specifically designed to evaluate whether the backdoor can persist long after poisoning has stopped.

As shown in Fig. S7, **FedAvg exhibits strong long-term vulnerability.** Consistent with the original Chameleon paper, FedAvg maintains a high ASR long after the attacker becomes inactive. With only 100 rounds of poisoning (10%, pixel "plus" trigger), FedAvg's ASR remains around **75% at round 800**, indicating a significant persistence of the backdoor.

**AUPF breaks Chameleon's persistence mechanism.** In contrast, AUPF prevents the backdoor from persisting. ASR shows a brief increase during the poisoning window but steadily diminishes once the attack ends. This suggests that AUPF prevents the attacker from embedding stable, convergence-resistant backdoor directions in the feature space, effectively removing the durability advantage that characterizes Chameleon.

**ACC stability.** The brief fluctuation in ACC around round 200 is expected, as Chameleon's poisoned optimization dynamics differ from benign training under our *from scratch* setup. After this transient period, AUPF stabilizes and maintains accuracy for the remainder of the training.

Overall, these results indicate that **FedAvg is highly vulnerable to persistent backdoor attacks**, while **AUPF suppresses the attacker's ability to maintain long-lasting backdoor effects**.

## A.9 Additional Comparison with a Robust Aggregation-Based Defense (AlignIns (Xu et al., 2025))

In this section, we further compare AUPF with the recently proposed server-side robust aggregation-based defense AlignIns (Xu et al., 2025).

We first compare AlignIns and AUPF on CIFAR-10 under the same settings as Table 1 of our submission. While AlignIns maintains relatively high clean accuracy, it **fails to suppress ASR** across all four attacks.

Table S6: Performance comparison in terms of ACC/ASR (%) between AlignIns and AUPF on CIFAR-10 across four attacks.

| Attack | BadNet | DBA | Neurotoxin | LP |
|---|---|---|---|---|
| **AlignIns** | 87.93/99.34 | 88.77/88.47 | 87.13/99.75 | 88.73/88.98 |
| **AUPF** | **84.61/4.96** | **85.66/2.03** | **84.49/4.80** | **84.97/8.00** |

As discussed in Appendix A.3, AlignIns is a **server-side robust aggregation-based defense** that detects malicious client updates prior to aggregation. Using the same analysis protocol outlined in Appendix A.3, we evaluated AlignIns under varying levels of data heterogeneity and different poison ratios, both of which are critical factors that influence the performance of robust aggregation methods. Specifically, we consider four settings: 1) IID, poison ratio = 0.5; 2) IID, poison ratio = 0.2;

3) non-IID, poison ratio = 0.5; 4) non-IID, poison ratio = 0.2. Using BadNet on CIFAR-10, we obtain the following results:

Table S7: FNR and ASR of AlignIns under different data heterogeneity and poison ratios (PR) on CIFAR-10 (BadNet).

| Setting | IID, PR=0.5 | IID, PR=0.2 | non-IID, PR=0.5 | non-IID, PR=0.2 |
|---|---|---|---|---|
| FNR (%) | 0.00 | 0.17 | 12.50 | 29.17 |
| ASR (%) | 0.81 | 0.75 | 78.40 | 99.34 |

These results demonstrate that AlignIns performs well when the data are IID and the poison ratio is high, as malicious updates can be more easily detected in such scenarios. However, its detection ability rapidly degrades when the **data become non-IID** and the **poison ratio decreases**, making benign and malicious updates much harder to distinguish. This corresponds exactly to the realistic FL scenarios in which robust aggregation methods typically break down.

Overall, the results confirm that while AlignIns is effective at detecting malicious updates in ideal scenarios, **it struggles when the poison ratio is small and the data distribution is non-IID**. In contrast, AUPF consistently achieves low ASR across all attacks by performing **local adversarial unlearning** on benign clients, which does not rely on separability between benign and malicious updates. We will include a short discussion in the revision to clarify these differences.

### A.10 SCALABILITY TO LARGER MODELS: RESNET-50

To assess the scalability of AUPF to larger models, we further evaluate it on a substantially deeper architecture, **ResNet-50**, on CIFAR-10 under the BadNet attack. The resulting performance (ACC% / ASR%) is reported in Table S8.

Table S8: Performance (ACC/ASR) (%) of different defense methods on CIFAR-10 using ResNet-50.

| FedAvg | Snowball⊖ | FDCR | Lockdown | AUPF |
|---|---|---|---|---|
| 89.70 / 98.83 | 84.76 / 99.99 | 89.23 / 99.99 | 85.51 /12.83 | **87.19 / 12.36** |

AUPF retains strong robustness (12.36% ASR) while maintaining competitive ACC. This demonstrates that AUPF remains effective on deeper and more expressive models where several existing defenses struggle.

**Computation overhead.** The runtime per benign client per round is shown in Table S9.

Table S9: Average per-round training time (in seconds) for benign clients.

| FedAvg | Lockdown | AUPF |
|---|---|---|
| 4.97s | 6.51s | 11.97s |

AUPF incurs a 2.4× slowdown relative to FedAvg on ResNet-50. This overhead is expected: AUPF performs a lightweight inner optimization step for unlearning, while Lockdown performs multi-step annealing and still yields substantially higher ASR. Importantly, the overhead grows **linearly with model depth** and does not exhibit superlinear scaling, indicating that AUPF remains tractable for modern FL models.

**Communication overhead.** AUPF does *not* modify model architecture or increase parameter dimensionality. Therefore, the transmitted update size remains **identical to FedAvg**. AUPF's additional computation occurs entirely locally and does not introduce extra communication rounds or additional payload. We note that some defenses, such as Lockdown, theoretically claim potential communication reduction via subspace masking. However, in practical FL implementations, the client must transmit both the masked model update *and* the corresponding binary mask. Consequently, the

effective communication payload is not reduced in practice and may even exceed FedAvg when the activated subspace ratio is above 50%.

In summary, AUPF scales to large models, introduces no additional communication cost, and its per-round computational overhead is a controlled linear factor that yields substantially stronger robustness than existing defenses.

Table S10: Scaling to larger client populations under 25% participation (ACC% / ASR%).

| Number of Clients | FedAvg | Snowball⊖ | FDCR | Lockdown | AUPF |
|---|---|---|---|---|---|
| 150 | 87.16/99.99 | 75.09/3.49 | 86.91/99.71 | 78.99/82.74 | 80.37/5.88 |
| 200 | 86.30/99.92 | 71.70/99.57 | 85.91/99.69 | 61.52/71.71 | 78.32/6.59 |

## A.11 SCALABILITY TO A LARGER NUMBER OF CLIENTS

Our method primarily targets defenses in the cross-device FL setting, which is characterized by partial client participation and large client populations. All our experiments already use a realistic 25% participation rate. Appendix Fig. S3 reports results under 40 and 100 clients. We further evaluate **150** and **200** clients under the same setting. As shown in Table S10, AUPF maintains stable ACC and consistently low ASR across all client scales.

AUPF exhibits consistent robustness as the client population increases, and its **local-only design** ensures that both communication and computation remain independent of the total number of devices. This makes AUPF naturally scalable to large-population cross-device FL deployments.

# B ADDITIONAL SETUP DETAILS

## B.1 FURTHER DETAILS ON LOCAL TRAINING OF MALICIOUS CLIENTS

The local optimization performed by a malicious client is formulated as

$$
\mathbf{w}_i^t = \arg\min_{\mathbf{w}} \left\{ \frac{1}{|\widehat{\mathcal{D}}_i|} \sum_{(\mathbf{x}_j^i, y_j^i) \in \widehat{\mathcal{D}}_i} L\big(f_{\mathbf{w}}(\mathbf{x}_j^i), y_j^i\big) + \frac{1}{|\widetilde{\mathcal{D}}_i|} \sum_{(\widetilde{\mathbf{x}}_j^i, \widetilde{y}) \in \widetilde{\mathcal{D}}_i} L\big(f_{\mathbf{w}}(\widetilde{\mathbf{x}}_j^i), \widetilde{y}\big) \right.
$$

$$
\left. + \frac{\lambda_1}{|\widetilde{\mathcal{D}}_i|} \sum_{(\widetilde{\mathbf{x}}_j^i, \widetilde{y}) \in \widetilde{\mathcal{D}}_i} L\big(f_{\mathbf{w}}(\widetilde{\mathbf{x}}_j^i \oplus \mathbf{\Delta}), \widetilde{y}\big) + \frac{\lambda_2}{|\widetilde{\mathcal{D}}_i|} \sum_{(\widetilde{\mathbf{x}}_j^i, \widetilde{y}) \in \widetilde{\mathcal{D}}_i} \big\| f_{\mathbf{w}}^e(\widetilde{\mathbf{x}}_j^i) - f_{\mathbf{w}}^e(\widetilde{\mathbf{x}}_j^i \oplus \mathbf{\Delta}) \big\|^2 \right\} \quad \text{(S1)}
$$

$$
\text{s.t.} \quad \mathbf{\Delta} = \arg\max_{\|\mathbf{\Delta}\| \le \rho} \frac{1}{|\mathcal{D}_i|} \sum_{(\mathbf{x}, y) \in \mathcal{D}_i} L\big(f_{\mathbf{w}_i^t}(\mathbf{x} \oplus \mathbf{\Delta}), y\big).
$$

Here, $\widehat{\mathcal{D}}_i$ denotes the (local) clean data and $\widetilde{\mathcal{D}}_i$ denotes the poisoned subset used by the attacker. The four terms in the objective correspond, respectively, to: 1) the standard classification loss on the clean local data, 2) the classification loss on poisoned examples, 3) an adversarial (or hardening) loss that enforces correct classification of poisoned inputs under the universal perturbation $\mathbf{\Delta}$ (weighted by $\lambda_1$), and 4) a feature-consistency regularizer (weighted by $\lambda_2$) that aligns the encoder outputs $f_{\mathbf{w}}^e(\cdot)$ of poisoned examples before and after perturbation.

## B.2 ATTACK METHODS

In this section, we provide a brief overview of the federated learning backdoor attack methods evaluated in our experiments.

- BadNet (Gu et al., 2017): BadNet is one of the most seminal backdoor attacks on deep neural networks, originally proposed in the centralized learning setting. In this attack, the adversary modifies a subset of the training data by embedding a predefined trigger into selected samples and relabeling them to a target class (i.e., the target label). Consequently, the trained model maintains high accuracy on clean inputs while consistently misclassifying trigger-embedded inputs as the target class. In federated learning, BadNet is adapted by allowing malicious clients to poison

a portion of their local training data through trigger injection and label manipulation, thereby implanting the backdoor into the global model during collaborative training.

- DBA (Distributed Backdoor Attack) (Xie et al., 2020): Unlike BadNet, DBA is specifically designed for the federated learning setting. To improve the persistence and stealthiness of the attack, (Xie et al., 2020) propose decomposing the global trigger pattern into distinct local patterns, with each malicious client embedding a portion of the trigger into its local poisoned data. Notably, when only one malicious client is present, DBA reduces to a standard backdoor attack. During inference, the full trigger pattern is embedded into each test sample to activate the backdoor.

- Neurotoxin (Zhang et al., 2022b): Neurotoxin is designed to improve the durability of backdoor attacks in federated learning, particularly under the single-shot attack setting. (Zhang et al., 2022b) observe that the backdoor effect tends to fade from the global model once the attacker ceases participation in training. To address this, Neurotoxin leverages the sparse nature of stochastic gradient descent (SGD) by projecting the adversarial gradient onto a subspace that is infrequently utilized by benign clients, thereby preserving the backdoor effect across subsequent training rounds. To identify coordinates that are rarely updated by benign clients, *Neurotoxin assumes access to the global gradient from the previous round on the server*, using it to approximate the benign gradient in the upcoming round. However, *this assumption may not hold in realistic federated learning scenarios*, potentially limiting the practicality of the attack.

- Layer-wise Poisoning (LP) Zhuang et al. (2024): This method introduces Layer Substitution Analysis to identify backdoor-critical (BC) layers, the small subset of layers whose parameters alone suffice to trigger a backdoor, by iteratively substituting layers between a benign model and a malicious model and measuring the resulting backdoor success rate. Specifically, LP selectively poisons only the BC layers and adapts the selection or scale of targeted layers to remain stealthy under aggregation filters, thereby enabling highly effective yet stealthy backdoors.

It is worth emphasizing that BadNet and DBA implant triggers solely by modifying raw training samples on malicious clients, whereas Neurotoxin and LP are more sophisticated attacks that also manipulate the training procedure, resulting in more stealthy backdoors.

## B.3 DEFENSE METHODS

In the following, we present a concise overview of the defense methods compared in our experiments.

- FoolsGold (Fung et al., 2020) is a defense mechanism designed to protect federated learning systems from Sybil based backdoor attacks, where adversaries create multiple fake clients (Sybils) that submit similar gradient updates to poison the global model. To detect such malicious clients, FoolsGold analyzes the similarity of client updates over time. It assigns adaptive learning rates to clients based on the diversity of their updates, thereby reducing the influence of those exhibiting highly similar gradient patterns, which are indicative of Sybil based backdoor attack.

- RLR (Özdayi et al., 2021): is a lightweight defense mechanism designed to mitigate backdoor attacks in federated learning. It is based on the observation that malicious clients often generate updates with sign patterns that deviate from those of benign clients, particularly in certain dimensions. To counter this, RLR adaptively reduces the server's learning rate for dimensions exhibiting inconsistent signs across client updates. This strategy effectively limits the impact of potential backdoor contributions without requiring modifications to the federated learning protocol or incurring significant computational overhead. However, the underlying assumption becomes more challenging to hold in non-IID settings, where natural variations in client updates may resemble malicious behavior. *In our experiments on the CIFAR-10 dataset, RLR performs worse than reported in the original paper (Özdayi et al., 2021). We attribute this discrepancy to differences in the experimental setup. While (Özdayi et al., 2021) evaluates RLR under full client participation with all 40 clients involved in each round, our experiments adopt partial client participation, selecting only 10 clients per round. As a result, the statistics used by RLR become less reliable.*

- RFA (Pillutla et al., 2022) is a robust aggregation method designed to enhance the resilience of federated learning systems against adversarial or corrupted client updates. Specifically, it employs the geometric median as the aggregation function, which is less sensitive to outliers. This approach ensures that the aggregated model remains robust even when a fraction of the participating clients provide compromised updates.

- Multi-Krum (Blanchard et al., 2017) is an extension of the Krum algorithm, designed to improve robustness against backdoor attacks in federated learning systems. While Krum selects a single client whose gradient update is closest to others, Multi-Krum generalizes this approach by selecting

multiple such updates. Specifically, it identifies the top $(N - F)$ gradient updates, where $F$ is the estimated number of malicious clients, that have the smallest sum of squared distances to their nearest neighbors. These selected updates are then aggregated by the server, while the rest are excluded, thereby mitigating the impact of malicious contributions.

- DeepSight (Rieger et al., 2022) is designed to address the shortcomings of traditional outlier-based filtering defenses, which often misclassify benign client updates as malicious, particularly in non-IID data settings. It adopts a two layer defense strategy: the first layer performs deep inspection of model updates using techniques such as Division Differences, Normalized Update Energies, and Threshold Exceedings to detect and filter out highly impactful poisoned models. The second layer applies weight clipping to further reduce the influence of any remaining malicious updates, thereby enhancing the robustness of federated learning systems against backdoor attacks.

- FLAME (Nguyen et al., 2022) is a defense framework designed to mitigate backdoor attacks in federated learning by estimating and injecting an appropriate amount of noise into the global model to eliminate backdoor effects. It integrates a threefold strategy: 1) automated clustering is employed to identify and exclude anomalous client updates; 2) adaptive weight clipping is applied to constrain the influence of potentially malicious updates; and 3) calibrated differential privacy noise is added to suppress any residual backdoor effects.

- Multi-metrics (Huang et al., 2023a) is a distance based defense strategy designed to detect backdoor attacks in federated learning systems. Traditional methods often rely on Euclidean distance to identify malicious client updates; however, this metric becomes less effective in high dimensional spaces and under non IID conditions. To address these limitations, Multi-metrics combines multiple distance measures, such as Manhattan and cosine distances, with dynamically adjusted weights to enable a more robust assessment of similarity among client updates. This adaptive approach improves the detection of malicious updates that closely resemble benign ones, thereby enhancing the system's resilience against stealthy backdoor attacks. In our experiments, Multi-metrics does not perform as well as expected. *This may be attributed to the original implementation (Huang et al., 2023a), which assigns a smaller learning rate and more local epochs to malicious clients, and a larger learning rate with fewer local epochs to benign clients, making their updates more distinguishable. However, this setting is unrealistic in practical scenarios, as the defense cannot control the behavior of malicious clients. In contrast, our experiments use the same learning rate and number of local epochs for all clients.*

- Snowball (Qin et al., 2024) is a defense framework that identifies trustworthy model updates through a two-stage election process. In the first stage, a bottom-up election is conducted, where each client votes for peer updates it considers reliable. In the second stage, a top-down election expands the trusted set by incorporating additional updates that exhibit high similarity, as measured by a variational autoencoder. This bidirectional mechanism enables robust aggregation by effectively filtering out potentially malicious contributions. Following the original Snowball paper (Qin et al., 2024), we also consider a variant, denoted Snowball⊖, which retains only the bottom-up election mechanism from the full Snowball framework. By selecting fewer clients during aggregation, Snowball⊖ typically achieves stronger backdoor defense compared to the full Snowball method.

- FDCR (Huang et al., 2024) is a defense mechanism designed for heterogeneous federated learning. It builds on the key observation that benign and malicious clients often exhibit distinct parameter importance degrees due to differences in their local data distributions. FDCR employs the Fisher Information Matrix (FIM) to quantify the importance of each parameter in local models and uses this information to reweight client gradient updates. This makes the updates of benign and malicious clients more distinguishable. Based on the reweighted gradients, FDCR applies unsupervised clustering to identify clients with significant gradient discrepancies as potential adversaries and excludes them from aggregation. For the remaining clients, FDCR further rescales each parameter according to the FIM, ensuring that more important elements are emphasized during aggregation.

- FLIP (Zhang et al., 2023b) proposes a trigger inversion strategy on benign clients to defend against backdoor attacks in federated learning. Specifically, it estimates potential backdoor triggers and uses them to augment local training data, enabling adversarial training that strengthens the robustness of local models. While effective under certain scenarios, FLIP has notable limitations: 1) it assumes that backdoor attacks occur only after the global model has converged, making it ineffective against continuous or persistent attacks; 2) its computational cost grows linearly with the number of class labels, presenting scalability issues for datasets with a large label space.

- Lockdown (Huang et al., 2023b) extends pruning-based backdoor defense techniques from centralized machine learning to the federated learning setting. The authors empirically observe that, in federated scenarios, backdoor-related parameters are often entangled with benign ones, making

direct pruning ineffective. To mitigate this, Lockdown constrains each client to perform local training within an isolated subspace that captures parameters important to its local data. These subspaces are derived through subspace pruning and subspace recovery. By enforcing disjoint parameter subspaces across clients, Lockdown effectively separates malicious updates from benign ones, enabling the reliable removal of backdoor parameters through majority voting during model aggregation.

## C  NOVELTY AND DISTINCTION FROM PRIOR ADVERSARIAL TRAINING

AUPF is fundamentally different from standard adversarial training. Rather than expanding robustness margins around clean samples, AUPF introduces an explicit *unlearning* objective that contracts poisoned feature directions and removes backdoor-sensitive associations. In addition, AUPF optimizes universal perturbations under non-IID local data and performs feature-space consistency alignment—capabilities that do not appear in conventional adversarial training.

AUPF contributes both a **new objective formulation** and a **non-IID–aware client-side design** that, to the best of our knowledge, have not been explored in prior FL backdoor defenses.

- First, AUPF incorporates an **explicit unlearning objective** rather than generic adversarial regularization. In Eq. (6), the feature-level term
$$\|f_{\mathbf{w}}^e(\mathbf{x}_j^i) - f_{\mathbf{w}}^e(\mathbf{x}_j^i \oplus \boldsymbol{\Delta})\|^2$$
is designed to **contract representations along poisoned directions** by enforcing consistency between clean and adversarial features. This mechanism is **conceptually distinct from adversarial training**, which increases decision margins but does not attempt to remove poisoned associations.
- Second, AUPF is, to the best of our knowledge, the **first client-side defense that explicitly optimizes universal perturbations under non-IID local data**. Although prior UAP-based defenses (e.g., FLIP) evaluate under non-IID settings, their algorithms are **not designed to address heterogeneous client distributions**. In contrast, AUPF leverages **prediction-level unlearning** to reveal poisoned-sensitive directions on each client and **feature-level alignment** to propagate this robustness through FedAvg.
- Third, AUPF explicitly targets **poisoned-direction removal**, not merely adversarial invariance. The proposed formulation suppresses backdoor-sensitive representational directions so they cannot survive aggregation across rounds. As demonstrated in Appendix A.6, removing the unlearning term restores poisoned feature clusters, confirming that AUPF performs **genuine unlearning rather than repackaged adversarial training**.

### C.1  COMPARISON WITH (WEI ET AL., 2023) AND (ZENG ET AL., 2022)

We summarize the key distinctions between AUPF and the approaches proposed in (Wei et al., 2023) and (Zeng et al., 2022) as follows.

**1) Setting Differences:** Both (Wei et al., 2023) and (Zeng et al., 2022) focus on **post-training defenses**, where they assume the model is already trained, and poisoned updates are removed after training. These methods typically rely on **a small number of clean samples** to mitigate backdoor effects. In contrast, **AUPF addresses a much more challenging federated in-training defense**. Specifically, in FL, malicious clients **continuously inject poisoned updates throughout training**, with **non-IID data** from different clients, and only a **subset of clients** is available for training in each round. Furthermore, **benign clients** have no access to global data, making this a much more dynamic and distributed problem. These differences in threat model imply that **post-training defenses** like those in (Wei et al., 2023) and (Zeng et al., 2022) cannot be applied or adapted to handle the **continuous poisoning** that occurs during FL optimization.

**2) Unlearning Mechanism:** The unlearning mechanism in (Wei et al., 2023) and (Zeng et al., 2022) is similar to traditional **adversarial training**, where adversarial perturbations are generated to create adversarial samples, and the model is then forced to classify them correctly. This approach is essentially a form of **margin-based adversarial training**. However, **AUPF introduces a novel perturbation-induced representation-unlearning mechanism**. Instead of using adversarial perturbations for robustness, **AUPF generates perturbations to explicitly activate poisoned-feature directions** inherited from the global model under non-IID data. This approach is fundamentally

different from traditional adversarial training or the methods in (Wei et al., 2023) and (Zeng et al., 2022), as **AUPF targets specific poisoned features** and uses a **feature-alignment strategy** to suppress those features, followed by propagating the resulting robustness to the global model.

In our approach, the **inner loop** uses universal adversarial perturbations (UAP) to activate the poisoned feature directions, which **explicitly target the poisoned features** injected by malicious clients. The **outer loop** performs **feature-space alignment**, conditioning on these activated poisoned features. This coupled process ensures that UAP exposes poisoned-feature vulnerabilities, alignment suppresses these activated features, and the resulting robustness is propagated to the global model.

This **coupled design** of local unlearning followed by global robustness propagation is crucial for federated learning with **non-IID data** and **continuous poisoning**. Without this mechanism, the **ASR** could remain high, as shown in Appendix A.6, where we provide mechanistic evidence demonstrating that the unlearning design is key to mitigating backdoors in FL settings.

**3) Conceptual and Algorithmic Novelty:** Therefore, AUPF's novelty lies in **its dynamic, in-training defense mechanism** that addresses continuous poisoning and **poisoned-feature unlearning** under **non-IID data**. This is a **conceptual and algorithmic advancement** beyond combining universal perturbations with feature alignment, as it enables the local model's robustness to be propagated to the global model, while directly addressing the ongoing poisoning throughout FL training.

In summary, **AUPF introduces clear conceptual and algorithmic novelty** that cannot be reduced to simply combining universal perturbations with feature alignment, nor can it be directly compared to post-training adversarial unlearning methods like those in (Wei et al., 2023) and (Zeng et al., 2022).

