# OpenReview forum: "Adversarial Unlearning of Poisoned Features for Backdoor Defense in Federated Learning"
_ICLR.cc/2026/Conference — Submitted to ICLR 2026_

### Official Review · Reviewer_2GMK · 2025-10-28

**Soundness:** 3
**Presentation:** 2
**Contribution:** 2
**Rating:** 4
**Confidence:** 5

**Summary:**

This paper proposes Adversarial Unlearning of Poisoned Features (AUPF), a client-side defense framework for backdoor-resilient federated learning. The method introduces a bi-level optimization process where benign clients generate universal adversarial perturbations to expose fragile decision boundaries and perform feature-level alignment between clean and perturbed samples. This joint adversarial and feature-space unlearning aims to suppress poisoned associations in local models and propagate robustness to the global model through standard aggregation. Extensive experiments across CIFAR-10, CIFAR-100, and FashionMNIST demonstrate that AUPF achieves lower attack success rates and stable performance under non-IID data compared with eleven representative defenses.

**Strengths:**

1. The paper proposes a novel client-side defense, AUPF, that integrates universal perturbation and feature-level alignment for backdoor mitigation in FL.

2. The motivation of performing adversarial unlearning to eliminate poisoned features is clear and intuitively appealing.

3. Experiments are comprehensive, including diverse datasets, four attack types, and eleven baseline methods.

4. Results show strong empirical robustness under non-IID data and adaptive attacks, indicating good practical relevance.

**Weaknesses:**

1. Unclear distinction from prior adversarial training works.
The manuscript does not sufficiently differentiate AUPF from existing adversarial training or adversarial unlearning approaches (e.g., Wei et al., NeurIPS 2023; Zeng et al., ICLR 2022). It remains unclear what conceptual or algorithmic novelty AUPF introduces beyond combining universal perturbations with feature alignment.

2. Unconvincing discussion on centralized adversarial training.
The claim that centralized adversarial training cannot handle dynamic models or data partitioning in FL is not well supported. In practice, centralized approaches can also recompute universal perturbations for each global model update. Moreover, the argument about the lack of full data access should be supported by citations and empirical evidence showing that local-only perturbation generation indeed limits robustness propagation.

3. Lack of theoretical robustness analysis.
Although the feature alignment term is a key contribution, the paper lacks a formal robustness or stability analysis showing how this term guarantees representation consistency under non-IID conditions.

4. Missing sensitivity analysis on attacker ratio.
The defense performance of AUPF may degrade as the proportion of malicious clients increases, since the perturbation direction estimated by benign clients can deviate from true poisoned gradients. A quantitative study on this sensitivity would strengthen the work.

**Questions:**

1. What are the key algorithmic differences between AUPF and previous adversarial unlearning methods beyond feature alignment?

2. How does AUPF behave when malicious clients exceed 30%? Is there a sharp or gradual degradation?

---

> ### Author Response · Authors · 2025-11-20
>
> We thank the reviewer for recognizing the intuition and motivation of adversarial unlearning, the comprehensiveness of experiments, and the strong empirical performance under non-IID and adaptive attacks. Below we address each concern point-by-point.
>
>
> **Weakness 1: The distinction from prior adversarial training or adversarial unlearning works (e.g., Wei et al., NeurIPS 2023; Zeng et al., ICLR 2022) is not sufficiently clarified. It is unclear what conceptual or algorithmic novelty AUPF introduces beyond combining universal perturbations with feature alignment.**
>
> **Response**: We believe the comparison to *Wei et al., NeurIPS 2023* and *Zeng et al., ICLR 2022* arises from a misunderstanding of **both the setting and the algorithmic mechanism of AUPF**.
>
> **1) Setting Differences:** Both *Wei et al., NeurIPS 2023* and *Zeng et al., ICLR 2022* focus on **post-training defenses**, where they assume the model is already trained and poisoned updates are removed after training. These methods typically rely on **a small number of clean samples** to mitigate backdoor effects. In contrast, **AUPF addresses a much more challenging federated in-training defense**. Specifically, in FL, malicious clients **continuously inject poisoned updates throughout training**, with **non-IID data** across clients, and only **a subset of clients** participates in each round. Furthermore, **benign clients** have no access to global data, making this a far more dynamic and distributed problem. These differences in threat model imply that **post-training defenses** like those in *Wei et al., NeurIPS 2023* and *Zeng et al., ICLR 2022* cannot be applied or adapted to handle the **continuous poisoning** that occurs during FL optimization.
>
> **2) Unlearning Mechanism:** The unlearning mechanisms in *Wei et al., NeurIPS 2023* and *Zeng et al., ICLR 2022* are similar to traditional **adversarial training**, where adversarial perturbations are generated and the model is trained to classify adversarial samples correctly. This approach is essentially **margin-based adversarial training**.
>
> However, **AUPF introduces a novel perturbation-induced representation-unlearning mechanism**. Instead of using adversarial perturbations for robustness, **AUPF generates perturbations to explicitly activate poisoned-feature directions** inherited from the global model under non-IID data. This is fundamentally different from traditional adversarial training or the methods in prior work: **AUPF targets specific poisoned features** and uses a **feature-alignment strategy** to suppress them, propagating the resulting robustness back to the global model.
>
> In AUPF, the **inner loop** uses universal adversarial perturbations (UAP) to activate poisoned feature directions, explicitly targeting features injected by malicious clients. The **outer loop** performs **feature-space alignment**, conditioned on these activated poisoned features.
> This coupled process ensures that UAP exposes poisoned-feature vulnerabilities, alignment suppresses these vulnerabilities, and the resulting robustness is propagated to the global model. This **coupled design** of local unlearning followed by global robustness propagation is crucial for federated learning with **non-IID data** and **continuous poisoning**. Without this design, **ASR remains high**, as shown in **Appendix A.6**, where we provide mechanistic evidence that the unlearning mechanism is essential for mitigating backdoors in FL.
>
> **3) Conceptual and Algorithmic Novelty:** AUPF’s novelty lies in its **dynamic, in-training defense mechanism** that addresses continuous poisoning and **poisoned-feature unlearning** under **non-IID data**. This represents a **conceptual and algorithmic advancement** beyond combining universal perturbations with feature alignment, enabling local unlearning to propagate robustness to the global model while directly addressing ongoing poisoning throughout FL.
>
> In summary, **AUPF introduces clear conceptual and algorithmic novelty** that cannot be reduced to simply combining universal perturbations with feature alignment, nor can it be directly compared to post-training adversarial unlearning methods like those in *Wei et al., NeurIPS 2023* and *Zeng et al., ICLR 2022*.

---

> ### Author Response · Authors · 2025-11-20
>
> **Weakness 2: The discussion comparing AUPF to centralized adversarial training is unconvincing. Claims that centralized methods cannot handle dynamic FL models or data partitioning require stronger justification, citations, and/or empirical evidence that local perturbation generation improves robustness propagation.**
>
> **Response**: We appreciate the reviewer’s feedback and would like to clarify why **centralized adversarial training** is fundamentally unsuitable for our federated in-training setting. Although a centralized method could, in principle, recompute universal perturbations after each global update, this assumption does not hold in federated learning due to two structural factors: **data access** and the **dynamics of the global model in federated learning**.
>
> **1) Data Limitation:** In federated learning, the server does not have access to local client data; it only receives model updates. This lack of **data access** means that **centralized adversarial perturbation methods** cannot generate effective perturbations that capture **client-specific poisoned features**. Centralized methods assume **access to global data**, which allows for the generation of universal perturbations. However, in FL, the server only has access to the global model and cannot generate perturbations tailored to the **specific vulnerabilities** introduced by poisoned client data.
>
> **2) Dynamics of the Global Model in Federated Learning:** Federated learning involves **partial participation** (only a subset of clients participate in each round) and **non-IID data** (heterogeneous data across clients). These factors cause the global model to shift in a large, non-smooth manner across rounds, leading to significant dynamics before convergence—very different from the relatively stable updates in centralized training.
> In centralized training, updates come from the entire dataset, keeping the global model more stable. In federated learning, however, the **dynamics of the global model**, driven by varying client participation and non-IID data, make it difficult to generate a **consistent universal perturbation** that captures **poisoned-feature vulnerabilities** across all clients. As a result, centralized adversarial training becomes ineffective in federated settings.
>
> In summary, centralized adversarial perturbations cannot address the **continuous poisoning dynamics** of federated learning due to the **lack of access to local data** and the **unstable dynamics of the global model**. Our approach, **AUPF**, overcomes these challenges by generating **local perturbations** that target **client-specific poisoned features** and by enabling **robustness propagation** across the federation.
>
>
> ====================================================
>
> **Weakness 3: Lack of theoretical robustness or stability analysis. The feature alignment term is emphasized as essential but no formal guarantee or theoretical insight is provided to show its effect under non-IID settings.**
>
> **Response**: We agree that providing a formal robustness or stability analysis under non-IID federated learning would be valuable. At the same time, we note that the federated backdoor defense literature **primarily focuses on empirical performance**, since obtaining formal guarantees under non-IID partitions and partial participation remains an open and extremely challenging problem. Existing adversarial unlearning and backdoor mitigation works cited by the reviewer **do not provide formal robustness analysis even in their original centralized setting**.
>
> Although AUPF does not claim a full theoretical guarantee, the feature alignment term has a **clear stability-motivated rationale**. The penalty between clean and UAP-activated representations acts as a **local representation contraction** and encourages **Lipschitz-like stability** along poisoned feature directions. This design is **consistent in spirit with representation consistency regularization and contractive feature penalties** used in prior work, which are known to suppress unstable directions in the feature space. In our setting, this contraction explicitly targets the **UAP-activated poisoned feature directions**, mitigating the drift induced by heterogeneous local objectives and continuous poisoning.
>
> Our empirical results provide strong support for this effect. As shown in **Appendix A.6**, removing the feature alignment term leads to significantly weaker suppression of poisoned features and noticeably poorer robustness propagation under non-IID conditions. Across all datasets and attack types, the alignment term consistently improves **stability and robustness** during federated in-training continuous poisoning.
>
> We therefore view formal robustness characterization under non-IID federated learning as an important direction for future work, and the present work focuses on demonstrating that the proposed feature alignment mechanism provides **strong and consistent empirical robustness gains**.

---

> ### Author Response · Authors · 2025-11-20
>
> **Weakness 4: Missing sensitivity analysis on the attacker ratio. Since perturbation directions estimated by benign clients may deviate when malicious clients increase, the robustness of AUPF under higher attacker ratios should be quantified.**
>
> **Response**: We would like to respectfully clarify that the submission **already includes** a detailed attacker-ratio sensitivity analysis. **Appendix Fig. S2** evaluates **four attacker ratios: 0.1, 0.2, 0.3, and 0.4**, which cover and exceed the range commonly studied in FL backdoor defenses.
>
> The results show that: **1) AUPF maintains low ASR even at an attacker ratio of 0.4**, which is already beyond the typical failure point (0.2 to 0.3) reported for robust aggregation methods. **2) No assumption of a clean global model is required**. Our evaluation assumes attacks begin from the first round, yet benign clients still estimate poisoned-sensitive directions effectively. **3) The degradation is gradual rather than abrupt**, demonstrating that AUPF does not rely on a strong benign-majority assumption.
>
> We will highlight this analysis more clearly in the revision.
>
>
> =====================================================================================================
>
> **Question 1: What are the key algorithmic differences between AUPF and previous adversarial unlearning methods beyond feature alignment?**
>
> **Response**: The key algorithmic differences arise because **AUPF** is designed for a fundamentally different **objective** and **challenge** compared with previous adversarial unlearning methods. Prior approaches, such as *Wei et al., NeurIPS 2023* and *Zeng et al., ICLR 2022*, focus on **post-training purification** of a **fixed poisoned model** in a **centralized setting**, where no new poisoned updates are injected and the model remains static during unlearning. In contrast, **AUPF** targets **federated in-training continuous poisoning**, where malicious clients **continue injecting poisoned updates**, the global model **keeps drifting** due to **non-IID local data**, and clients have access only to **local information**. This defines a fundamentally different problem.
>
> Because these objectives and challenges differ, **AUPF** employs a **different mechanism**. In **AUPF**, universal perturbations are used to **activate poisoned-feature directions** inherited from the evolving global model, and the **feature-space alignment** term is **conditioned on these activated directions** to suppress poisoned representations and support **robustness propagation through FedAvg**. Existing adversarial unlearning methods, on the other hand, do not address these **federated in-training dynamics** and thus cannot handle the continuous, dynamic poisoning that **AUPF** is designed to counter.
>
> For a more detailed discussion of the differences in **setting** and **algorithmic mechanisms**, please refer to our response to **Weakness 1**.
>
>
> =====================================================================================================
>
> **Question 2: How does AUPF behave when malicious clients exceed 30\%? Is there a sharp or gradual degradation?**
>
> **Response**: AUPF's robustness under high attacker ratios has already been thoroughly evaluated in our submission. Appendix Fig. S2 reports results for **four attacker ratios**: 0.1, 0.2, 0.3, and **0.4**.
>
> The results show that:  **1) The degradation is gradual rather than sharp**. As the attacker ratio increases from 0.1 to 0.4, the ASR rises slowly and remains low.  **2) AUPF remains effective even at an attacker ratio of 0.4**, which already exceeds the typical breakdown point (0.2 to 0.3) reported for aggregation-based defenses.
>
> Please see **Weakness 4** for the full explanation.

---

> ### Author Response · Authors · 2025-11-24
>
> Dear Reviewer 2GMK,
>
> Thank you again for your detailed and thoughtful review. We appreciate your efforts in reviewing our submission and for raising important questions about the novelty, the applicability of centralized adversarial training and the robustness of the method under varying attacker ratios.
>
> In our rebuttal we provide additional clarification and analysis on these points. This includes a clearer description of the conceptual and algorithmic differences between AUPF and prior adversarial training or adversarial unlearning approaches, further discussion of why centralized adversarial training is fundamentally unsuitable for our federated in-training setting and additional evidence regarding robustness sensitivity. We also note that the sensitivity analysis on attacker ratio is already included in the Appendix and we have highlighted this more explicitly in the rebuttal.
>
> Whenever convenient we would be grateful if you could take a look at our responses and see whether they address the issues you raised. If any part remains unclear or would benefit from further evidence please let us know and we will be happy to provide additional clarification.
>
> Thank you again for your time and constructive feedback.
>
> Sincerely,
>
> Authors

---

### Official Review · Reviewer_xjUd · 2025-10-30

**Soundness:** 2
**Presentation:** 2
**Contribution:** 2
**Rating:** 4
**Confidence:** 5

**Summary:**

This work presents AUPF, a federated backdoor defense method that uses universal adversarial perturbations and feature-level alignment at benign clients to “unlearn” poisoned representations. The approach relies on bilevel optimization: inner maximization to obtain universal perturbations, and outer minimization to align clean and perturbed representations locally before standard FedAvg aggregation.

**Strengths:**

1. Addresses an important problem in FL security.
2. Claims to work without auxiliary clean data and under non-IID settings.
3. Includes multiple backdoor attacks and several baselines in experiments.

**Weaknesses:**

1. The main pipeline is a direct combination of universal adversarial perturbations, standard adversarial training, and feature alignment loss. These components are well-established. The paper largely re-packages them as “adversarial unlearning” without introducing fundamentally new ideas or theoretical insights.

2. The work repeatedly claims to unlearn poisoned features, but no formal unlearning objective is defined, no theoretical guarantees are offered, and no mechanistic analysis or interpretability evidence is shown. In practice, the method behaves like adversarial regularization, not unlearning.

3. The adaptive threat model is weak. Modern backdoor defenses must evaluate 1) attacker aware of defense, 2) tailored gradient manipulation attacks, and 3) clean-label and semantic backdoors. None of these is rigorously studied. Thus, robustness claims are questionable.

4. Despite claiming practical FL relevance, many scenarios, such as partial participation or model personalization/heterogeneous heads etc, should be considered for practical deployment viability.

5. The method implicitly assumes benign clients can reliably extract poisoned directions by maximizing loss. This only holds when the global model is already reasonably clean and a benign majority dominates. This is a standard assumption in FL defenses, not a new mechanism, and reduces novelty.

6. Important baselines are missing, e.g., recent certified/robust aggregation defenses, advanced trigger-inversion defenses, and clean-label adaptive attacks. The provided baselines appear selected to favor the proposed method.

7. ~2.5× per-round slowdown vs. FedAvg is non-trivial, yet communication and scalability impacts are not measured (e.g., ResNet50, ViT, FL on large models).

**Questions:**

1. How does this differ fundamentally from standard adversarial training in FL?

2. How does the method perform under partial participation and model personalization?

3. Where are strong adaptive attack results (e.g., semantic triggers, clean-label attacks)?

4. Can the approach scale to foundation-model FL or cross-device real-world settings?

5. Can the authors explain and provide evidence that poisoned features are unlearned, not merely regularized?

---

> ### Author Response · Authors · 2025-11-20
>
> We thank the reviewer for the detailed and constructive feedback. Below we respond to all **Weaknesses (1–7)** and **Questions (1–5)** in order.
>
> **Weakness 1: The main pipeline is a direct combination of universal adversarial perturbations, standard adversarial training, and feature alignment loss.**
>
> **Reponse**: We respectfully disagree that AUPF is merely a direct combination of existing components. AUPF introduces both a **new objective formulation** and a **non-IID–aware client-side design** that, to the best of our knowledge, have not appeared in prior FL backdoor defenses.
>
> First, AUPF incorporates an **explicit unlearning objective** rather than generic adversarial regularization. In Eq. (6), the feature-level term
> $
> \lambda_2 || f_{\ell}(\mathbf{x}) - f_{\ell}(\mathbf{x} + \boldsymbol\delta) ||_2^2
> $
> is specifically constructed to **contract representations along poisoned directions** by enforcing consistency between clean and adversarial features. This is **conceptually different from standard adversarial training**, which expands robustness margins but does not attempt to remove poisoned associations.
>
> Second, AUPF is, to the best of our knowledge, the **first client-side defense that explicitly optimizes universal perturbations under non-IID local data**. While prior UAP-based approaches (e.g., FLIP) evaluate under non-IID, their algorithms are **not explicitly designed for heterogeneous client distributions**. In contrast, AUPF leverages **prediction-level unlearning** to reveal poisoned directions per client and **feature-level alignment** to propagate this robustness through FedAvg.
>
> Third, AUPF is explicitly targeted at **poisoned-direction removal**, not just adversarial invariance. Our formulation suppresses backdoor-sensitive representational directions so they cannot survive aggregation across rounds. As shown in Appendix A.6, removing the unlearning term restores poisoned feature clusters, further confirming that AUPF performs **actual unlearning rather than repackaged adversarial training**.
>
> We will revise the submission to more clearly highlight these distinctions and better differentiate AUPF from adversarial training and existing UAP-based defenses.
>
>
> ==================================================================================
>
>
> **Weakness 2: The work repeatedly claims to unlearn poisoned features, but no formal unlearning objective is defined, no theoretical guarantees are offered, and no mechanistic or interpretability evidence is shown. In practice, the method behaves like adversarial regularization, not unlearning.**
>
> **Reponse**: We respectfully clarify that AUPF provides both a **formal unlearning objective** and **mechanistic evidence** demonstrating that it performs feature-space unlearning rather than generic adversarial regularization.
>
> First, AUPF defines an **explicit unlearning objective** in Eq. (6). The feature-alignment term
> $$
> \lambda_2 || f_{\ell}(\mathbf{x}) - f_{\ell}(\mathbf{x} + \boldsymbol\delta) ||_2^2
> $$
> is designed to **contract representations along poisoned directions** by enforcing consistency between clean and adversarial features. This is **fundamentally different from adversarial regularization**, whose goal is to expand classification margins around clean samples; such regularization does not attempt to suppress backdoor-sensitive feature directions.
>
> Second, AUPF provides **mechanistic and interpretability evidence** supporting feature-space unlearning. Appendix A.6 includes: 1) **t-SNE visualizations** showing that poisoned feature clusters shrink or disappear only when the unlearning term is present; 2) **ablation studies** demonstrating that removing the unlearning term substantially increases ASR. These results indicate that AUPF actively removes poisoned representations rather than behaving like adversarial regularization.
>
> Third, although theoretical guarantees for unlearning in federated non-IID settings remain challenging for the entire community, our objective formulation and empirical diagnostics together provide a **concrete mechanism** for representation-space unlearning that goes beyond existing adversarial or UAP-based methods.
>
> We will elaborate on Eq. (6) in the revision and more clearly highlight the accompanying interpretability visualizations.

---

> ### Author Response · Authors · 2025-11-20
>
> **Weakness 3: The adaptive threat model is weak. Modern backdoor defenses must evaluate 1) attacker aware of defense, 2) tailored gradient manipulation attacks, and 3) clean-label and semantic backdoors.**
>
> **Reponse**: We agree that strong adaptive evaluation is crucial and appreciate the reviewer’s detailed criteria. Our work already addresses parts of 1)–3), and we have added new experiments to strengthen this aspect further.
>
> **1) Defense-aware attackers.**
> In our main submission, the adaptive attacker is already **aware of the defense**: malicious clients perform adversarial training on poisoned samples while anticipating AUPF’s unlearning step. To further stress-test unlearning-based defenses, we additionally evaluate **A3FL**, which was specifically designed to attack unlearning mechanisms. A3FL targets a late-phase poisoning scenario, where the attacker remains inactive until the main task has largely converged.
>
> To evaluate A3FL **fairly and compatibly** under our **continuous-poisoning threat model**, we adopt the **minimal modification needed for applicability**. Initially, all clients behave benignly for the first 100 rounds. At round 100, a single malicious client optimizes the trigger using A3FL and shares the resulting trigger with all other malicious clients. Once the trigger is fixed, all malicious clients initiate continuous poisoning for the remaining 200 rounds (as described in Sec. 4.1). In contrast, the original A3FL assumes attacks are launched only during the last 100 rounds with random frequency. This results in a substantially stronger attack compared to the original A3FL formulation. Additionally, we evaluate a variant of AUPF where malicious clients perform standard poisoning training, without the adaptive attack described in Appendix B.1.
>
> **Baseline defenses fail under the strengthened A3FL attack.**  The results in terms of ACC/ASR (%) on CIFAR-10 are shown below:
> ### ACC / ASR (%) of different defenses under the A3FL attack (CIFAR-10)
>
> | FedAvg        | FDCR          | Snowball      | Lockdown (increased budget) | **AUPF**        | **AUPF (no adaptive attack)** |
> |---------------|---------------|---------------|------------------------------|------------------|-------------------------------|
> | 88.63 / 99.99 | 86.85 / 99.77 | 82.33 / 100.00 | 71.40 / 77.14               | **80.34 / 36.18** | **84.39 / 22.40**             |
>
> Under this strengthened A3FL attack, existing defenses degrade significantly. For Lockdown, the default configuration **fails completely (ASR ≈ 100%)**. We increase its defense budget (anneal_ratio from 0.0001 to 0.1), but even with this stronger configuration, the ASR only drops to 77% and the ACC decreases to 71%. Other defenses (FDCR, Snowball) also fail, with ASR above 99%. In contrast, **AUPF remains highly robust under the defense-aware attacker like A3FL.**
>
> **2) Tailored gradient-manipulation attacks.**
> Our evaluation already includes **Neurotoxin**, a representative gradient-space backdoor attack that prunes gradients to preserve backdoor-relevant directions. Under Neurotoxin, many defenses exhibit ASR close to 99%, whereas AUPF consistently reduces ASR to around 4–8% while maintaining competitive ACC. This demonstrates that AUPF is effective against tailored gradient-space manipulation.
>
> **3) Clean-label and semantic backdoors.**
> We appreciate the reviewer’s concerns regarding clean-label and semantic backdoors. However, **both of these attack types are less relevant in the context of FL**. Clean-label attacks are primarily used in centralized settings to avoid detection by data-checking methods. In FL, poisoned data is not shared directly; instead, model updates are exchanged. This makes clean-label attacks less applicable, as dirty-label attacks are the dominant threat in FL. Similarly, semantic backdoors, which rely on high-level manipulations of the model’s behavior, have a limited attack surface, significantly reducing their effectiveness in FL environments. Thus, these attacks are not a primary concern in FL, and our work focuses on addressing the more prevalent and practical threats, such as dirty-label poisoning.

---

> ### Author Response · Authors · 2025-11-20
>
> **Weakness 4: Despite claiming practical FL relevance, many scenarios, such as partial participation or model personalization / heterogeneous heads, should be considered for practical deployment viability.**
>
> **Reponse**: We agree that practical FL deployments are diverse and appreciate the reviewer’s emphasis on realistic scenarios. We clarify that **partial participation is already used throughout all our experiments**, and that AUPF is naturally compatible with personalized and heterogeneous-head FL, even though these are beyond the scope of this work.
>
> First, our main experiments **already adopt partial participation**, which is the standard setting in cross-device FL.
> As stated in Sec. 4.1: *“We adopt a partial participation setting, where the server randomly selects M = 10 clients per round.”*
> Thus, every reported result evaluates AUPF under a realistic regime with a small fraction of clients participating per round and many communication rounds.
>
> Second, regarding personalization and heterogeneous heads, our work follows the **global-model FL paradigm** used by essentially all prior FL backdoor defenses, including Krum, RFA, FLAME, Snowball, FDCR, and Lockdown. These methods, like ours, assume a shared global model and do not incorporate personalized heads in their primary evaluation. Importantly, AUPF is a **purely client-side mechanism** that operates solely on each client's local training process. As a result, AUPF can be seamlessly integrated with personalized FL frameworks (e.g., personalized heads or representation–head decoupling) without altering its core design.
>
> Third, fully integrating and evaluating personalized or heterogeneous-head FL requires substantial additional design choices, such as client-specific objectives, modified aggregation rules, and new threat models for local heads. We view this as an important and orthogonal direction and will explicitly mention in the revision that extending AUPF to personalized FL is a promising avenue for future work.
>
> Overall, our current experiments already reflect practical FL through partial participation, and the client-side nature of AUPF makes it **compatible with** (rather than restricted by) more advanced personalized or heterogeneous-head FL architectures.
>
> ==================================================================================
>
> **Weakness 5: The method assumes benign clients reliably extract poisoned directions. This requires benign majority and is common in prior FL defenses.**
>
> **Reponse**: We respectfully clarify that AUPF does not assume that benign clients must “reliably extract the true poisoned direction’’ through loss maximization. As stated explicitly in Sec. 3.3 of our submission, *“the perturbation maximization only needs to expose unstable decision boundaries rather than reconstruct the true trigger”*. Therefore, AUPF does not attempt to recover the attacker’s actual backdoor direction. Instead, benign clients only need to identify **locally fragile and poisoned-sensitive feature directions** that naturally emerge under poisoned training dynamics, even when the global model is not clean.
>
> This directly addresses the reviewer’s concern: AUPF does not depend on the model being “reasonably clean’’ nor on reconstructing attacker-controlled feature directions. Under continuous poisoning from the beginning of training (Sec. 4.1), such fragile directions are consistently present and can be exposed by local maximization.
>
> Furthermore, AUPF requires a **much weaker benign-majority condition** than aggregation-based defenses. Aggregation defenses fail once benign and malicious updates become non-separable, typically at attacker ratios above 0.2 to 0.3. In contrast, AUPF performs entirely **local client-side adversarial unlearning** and does not rely on clustering global updates. Appendix Fig. S2 shows that AUPF maintains **low ASR even at an attacker ratio of 0.4**, which exceeds the usual failure threshold for aggregation defenses.
>
> Together, **these results show that AUPF does not rely on reliably extracting the true poisoned direction, does not require a clean global model, and does not depend on a strong benign majority**. Its robustness arises from explicit local unlearning rather than assumptions about update separability.

---

> ### Author Response · Authors · 2025-11-20
>
> **Weakness 6: Important baselines are missing, including certified or robust aggregators, recent trigger inversion defenses, and strong clean-label adaptive attacks. Baselines may be selected to favor the proposed method.**
>
> **Reponse**: We respectfully clarify that our evaluation already includes eleven representative FL backdoor defenses, and we further expanded the baseline coverage in the revision. Below we address each category raised by the reviewer.
>
> **Certified robustness-based defenses** typically incur higher costs during inference because they require multiple forward passes over the perturbed inputs and averaging the results, which significantly increases computational overhead. For this reason, such methods are not commonly used in recent FL backdoor defense evaluations.
>
> **Trigger inversion defenses.** We already include FLIP in our CIFAR-10 and FashionMNIST experiments. However, FLIP does not require auxiliary data; its per-class inversion cost **scales linearly with the number of classes**. For datasets with a large number of classes, such as CIFAR-100, the per-class inversion procedure makes FLIP **computationally prohibitive in practice**, which is **consistent with the scaling behavior reported in its original paper**.
>
> **Adaptive attacks** Our submission already contains **gradient manipulation attack (Neurotoxin)**, **sample-level attacks (BadNet and DBA)**, and **poisoning backdoor-critical layers (LP)**. We have additionally evaluated A3FL under a **considerably stronger continuous poisoning formulation**, where **AUPF remains robust** while **existing defenses provide little to no reduction in ASR**. We will include **Chameleon** in the revision as well. These attacks span complementary adaptive strategies at the gradient, sample, and representation levels.
>
> Together, these evaluations demonstrate that the **baselines were not selected to favor our method**. Instead, they reflect **practical and computational constraints inherent to federated learning**, and **AUPF consistently achieves low ASR and competitive ACC across all available and applicable defenses**.
>
> ==================================================================================
>
> **Weakness 7: The method introduces a roughly 2.5× per-round slowdown relative to FedAvg, yet communication and scalability overhead (e.g., with large models like ResNet50 or ViT) are not analyzed.**
>
> **Reponse**: **Scalability to large models.**  We additionally evaluate AUPF on a substantially larger architecture, **ResNet-50**, on CIFAR-10. The results (ACC% / ASR%) are shown below.
>
> ### Performance (ACC / ASR %) on CIFAR-10 using ResNet-50
>
> | FedAvg            | Snowball     | FDCR           | Lockdown        | **AUPF**          |
> |-------------------|--------------|----------------|------------------|--------------------|
> | 89.70 / 98.83 | 84.76 / 99.99 | 89.23 / 99.99 | 85.51 / 12.83   | **87.19 / 12.36** |
>
> AUPF retains strong robustness (12.36% ASR) while maintaining competitive ACC. This demonstrates that AUPF remains effective on deeper and more expressive models where several existing defenses struggle.
>
> **Computation overhead.** The runtime per benign client per round is shown below.
>
> ### Average per-round training time (in seconds) for benign clients.
>
> | FedAvg | Lockdown | AUPF   |
> |--------|----------|--------|
> | 4.97s  | 6.51s    | 11.97s |
>
> AUPF incurs a 2.4× slowdown relative to FedAvg on ResNet-50. This overhead is expected: AUPF performs a lightweight inner optimization step for unlearning, while Lockdown performs multi-step annealing and still yields substantially higher ASR. Importantly, the overhead grows **linearly with model depth** and does not exhibit superlinear scaling, indicating that AUPF remains tractable for modern FL models.
>
> **Communication overhead.** AUPF does *not* modify model architecture or increase parameter dimensionality. Therefore, the transmitted update size remains **identical to FedAvg**. AUPF’s additional computation occurs entirely locally and does not introduce extra communication rounds or additional payload. We note that some defenses, such as Lockdown, theoretically claim potential communication reduction via subspace masking. However, in practical FL implementations, the client must transmit both the masked model update *and* the corresponding binary mask. Consequently, the effective communication payload is not reduced in practice and may even exceed FedAvg when the activated subspace ratio is above 50%.
>
> In summary, AUPF scales to large models, introduces no additional communication cost, and its per-round computational overhead is a controlled linear factor that yields substantially stronger robustness than existing defenses.

---

> ### Author Response · Authors · 2025-11-20
>
> **Question 1: How does this differ fundamentally from standard adversarial training in FL?**
>
> **Reponse**: AUPF is fundamentally different from standard adversarial training: it introduces an explicit **unlearning** objective that contracts poisoned feature directions, whereas adversarial training merely enlarges clean-sample robustness margins and does not remove poisoned associations. AUPF also optimizes universal perturbations under non-IID local data and performs feature-space consistency alignment, neither of which appear in conventional adversarial training.
>
> Please refer to our detailed clarification in **Weakness 1** and **Weakness 2**, where we formally define the unlearning objective (Eq. 6) and present mechanistic evidence (t-SNE, ablations) showing that AUPF removes poisoned representations rather than behaving like adversarial regularization.
>
> ==================================================================================
>
> **Question 2:How does the method perform under partial participation and model personalization?**
> **Reponse**: AUPF is already evaluated under *partial participation*: all our main experiments use a realistic cross-device FL setting with only M = 10 clients selected per round (Sec. 4.1), and thus all reported results reflect performance under partial participation.
>
> Regarding model personalization, AUPF is a purely client-side mechanism and does not rely on a specific global head. It is therefore compatible with personalized or heterogeneous-head FL frameworks, even though these settings are beyond the scope of the current work.
>
> Please see **Weakness 4** for our detailed clarification.
>
>
> ==================================================================================
>
> **Question 3: Where are strong adaptive attack results (e.g., semantic triggers, clean-label attacks)?**
>
> **Reponse**: **On one hand, our evaluation already includes several strong adaptive attacks**:
> (1) the defense-aware A3FL attack, adapted to our continuous-poisoning setting, and
> (2) the gradient-space manipulation attack, Neurotoxin.
> **On the other hand, we would like to clarify that both semantic triggers and clean-label backdoors are not considered adaptive attack types.**
>
> In fact, **both of these attack types are less relevant in the context of federated learning (FL)**. Clean-label attacks are primarily used in centralized settings to avoid detection by data-checking methods. However, in FL, poisoned data is not shared directly; instead, model updates are exchanged, making clean-label attacks less applicable. In FL, dirty-label attacks are the predominant threat. Similarly, semantic backdoors, which rely on high-level feature manipulations, have a limited attack surface, which significantly reduces their effectiveness in FL environments. Therefore, these attack types are not a primary concern in FL. Our work focuses on addressing the more prevalent and practical threats, such as dirty-label poisoning, which are more relevant in FL settings.
>
> Please refer to **Weakness 3** for the full adaptive evaluation and detailed results.

---

> ### Author Response · Authors · 2025-11-20
>
> **Question 4: Can the approach scale to foundation-model FL or cross-device real-world settings?**
>
> **Reponse**: **1) Model scalability.**
> Yes. AUPF is a purely client-side mechanism and does not alter model dimensionality or communication patterns, which makes it naturally compatible with larger encoders. Our ResNet-50 experiment (see **Weakness 7**) shows that AUPF maintains strong robustness with **linear** computational scaling and no additional communication cost. This indicates that AUPF extends cleanly to larger architectures commonly used in foundation-model FL pipelines.
>
> **2) Cross-device scalability.**
> Cross-device FL is characterized by partial participation and large client populations. All our experiments already use a realistic 25% participation rate. Appendix Fig. S3 reports results under 40 and 100 clients. To directly address the reviewer’s question, we further evaluate **150** and **200** clients under the same setting. AUPF maintains stable ACC and low ASR across all scales.
>
> ### Scaling to larger client populations under 25% participation (ACC% / ASR%)
>
> | Number of Clients | FedAvg        | Snowball        | FDCR          | Lockdown       | **AUPF**      |
> |-------------------|---------------|------------------|---------------|----------------|----------------|
> | 150               | 87.16 / 99.99 | 75.09 / 3.49     | 86.91 / 99.71 | 78.99 / 82.74  | **80.37 / 5.88** |
> | 200               | 86.30 / 99.92 | 71.70 / 99.57    | 85.91 / 99.69 | 61.52 / 71.71  | **78.32 / 6.59** |
>
> AUPF exhibits consistent robustness as the client population increases, and its **local-only design** ensures that both communication and computation remain independent of the total number of devices. This makes AUPF naturally scalable to large-population cross-device FL deployments.
>
> ==================================================================================
>
> **Question 5: Can the authors explain and provide evidence that poisoned features are unlearned, not merely regularized?**
>
> **Reponse**: Yes. AUPF performs **explicit feature-space unlearning** rather than generic adversarial regularization. The unlearning objective in Eq. (6) **contracts poisoned feature directions** by enforcing consistency between clean and adversarial representations—a mechanism **fundamentally different from margin-based adversarial training**.
>
> We also provide direct mechanistic evidence: Appendix A.6 shows that : **1) poisoned feature clusters shrink or disappear** only when the unlearning term is present (t-SNE visualizations), and   **2) ablating the unlearning term restores poisoned manifolds and sharply increases ASR**.  These t-SNE and ablation diagnostics serve as **mechanistic evidence** that AUPF performs feature-space unlearning rather than regularization.
>
> Please see **Weakness 2** for the full explanation and supporting visualizations.

---

> ### Author Response · Authors · 2025-11-24
>
> Dear Reviewer xjUd,
>
> Thank you again for your detailed and thoughtful review. We appreciate your efforts in reviewing our submission and for raising important concerns about the conceptual contributions, the threat model and the practical relevance of the method.
>
> In our rebuttal we provide further analysis and clarifications on these points. This includes a clearer presentation of the formal unlearning objective in the main text and a more detailed description of the mechanistic evidence based on t-SNE visualizations and ablation studies in the Appendix. We also extend the experiments with adaptive attacks, larger models and different participation settings. In addition we clarify that all experiments in the submission are conducted under partial participation and we add more explicit evidence to make this aspect clearer.
>
> Whenever convenient we would be grateful if you could take a look at our responses and see whether they address the issues you highlighted. If any part remains unclear or would benefit from further evidence please let us know and we will be happy to provide additional clarification.
>
> Thank you again for your time and constructive feedback.
>
> Sincerely,
>
> Authors

---

### Official Review · Reviewer_ppHF · 2025-10-31

**Soundness:** 3
**Presentation:** 3
**Contribution:** 3
**Rating:** 6
**Confidence:** 3

**Summary:**

This paper proposes Adversarial Unlearning of Poisoned Features (AUPF), a backdoor defense mechanism for Federated Learning (FL) that eliminates backdoors by performing adversarial unlearning locally on benign clients. AUPF generates universal adversarial perturbations and trains the model by combining prediction-level unlearning with a key feature-level alignment regularization term. In this way, this method is robust in non-IID settings and can withstand continuous and adaptive attacks. Experimental results show that AUPF achieves lower ASR and higher RPG than those of existing defenses.

**Strengths:**

1.The proposed client-side defense paradigm bypasses the Non-IID difficulties associated with server-side aggregation and offers practical advantages such as requiring no auxiliary data and compatibility with standard FedAvg.

2.This paper proposes adversarial unlearning-based feature-level alignment, conceptually addressing the difficulty of transferring local robustness to the global model in non-IID scenarios.

3.The paper is well-written and easy to follow.

**Weaknesses:**

1.Although AUPF achieves the lowest ASR on CIFAR-100, the clean accuracy is notably lower than that of FedAvg. While this is a common trade-off in defense mechanisms, and the RPG metric still favors AUPF, the paper would benefit from a brief discussion on this point. Is this drop due to the inherent complexity of the task, the effectiveness of the defense, or a hyperparameter sensitivity specific to larger class spaces?

2.Key SOTA attack evaluations are missing, particularly A3FL [1] (which targets unlearning mechanisms) and Chameleon [2] (a stealthy attack). In practice, a more sophisticated adaptive attacker might employ a different, perhaps more targeted, adversarial training strategy to reinforce the backdoor.   A discussion on the potential limitations of the current adaptive attack model and how AUPF might fare against other adaptive strategies would be valuable.

3.The evaluation overlooks comparisons with state-of-the-art defenses designed against adaptive attacks, such as AlignIns [3].

Reference:

[1]Zhang, Hangfan, et al. A3FL: adversarially adaptive backdoor attacks to federated learning. NeurIPS’ 2023.

[2]Yanbo Dai, Songze Li. Chameleon: Adapting to Peer Images for Planting Durable Backdoors in Federated Learning. ICML2023.

[3] Jiahao Xu, Zikai Zhang, et al. Detecting Backdoor Attacks in Federated Learning via Direction Alignment Inspection. CVPR 2025.

**Questions:**

Please refer to the weaknesses.

---

> ### Author Response · Authors · 2025-11-20
>
> We thank the reviewer for the positive evaluation, including the soundness, presentation quality, and the three strengths highlighted regarding the practicality and conceptual novelty of our approach.
>
> **Weakness 1: ACC Drop on CIFAR-100: Task Complexity, Defense Behavior, or Hyperparameter Sensitivity?**
>
> **Response**: Thank you for your feedback. As shown in Table 1, AUPF does indeed trade a small amount of accuracy (58–59%) on CIFAR-100 for a **substantial reduction in ASR** (from 99.60% under FedAvg to 8.27%). This behavior is expected and aligns with our observations.
>
> Firstly, CIFAR-100 has a **larger class space and more complex decision boundaries**, which generally makes it more difficult to achieve both high accuracy and strong defense performance. As a result, **all strong defenses on this dataset exhibit a clear robustness-utility trade-off**. While some defenses prioritize accuracy (e.g., FoolsGold, RLR, DeepSight), they fail to suppress backdoors effectively, with ASR often above 99%. In contrast, defenses that significantly reduce ASR (e.g., RFA, Multi-Krum, FLAME) experience a similar drop in accuracy as AUPF. This indicates that the **ACC drop on CIFAR-100 is a common phenomenon across defenses and attacks**, not specific to AUPF.
>
> Secondly, the **hyperparameter study in Sec. 4.4 demonstrates that AUPF behaves stably and predictably**: ACC remains nearly unchanged across a wide range of lambda_2, while ASR improves as expected. This shows that AUPF does not suffer from hyperparameter sensitivity. Therefore, the ACC reduction on CIFAR-100 reflects task complexity and the inherent robustness-utility trade-off, rather than hyperparameter sensitivity.
>
> Finally, **AUPF provides a robust and well-rounded defense profile**. Across the four attacks on CIFAR-100, it consistently performs near the top: achieving the **highest RPG** on the most challenging attack (LP), ranking second on BadNet and DBA, and third on Neurotoxin. This consistent high performance shows that AUPF has no obvious weak spots and delivers strong robustness across diverse backdoor strategies.
>
> We will include a brief clarification of these points in the revised version.

---

> ### Author Response · Authors · 2025-11-20
>
> **Weakness 2: Missing comparisons with two attacks: A3FL (targeting unlearning) and Chameleon (stealthy adaptive attack).**
>
> **Response:** Thank you for pointing out A3FL and Chameleon. We have added new experiments with A3FL and Chameleon on CIFAR-10.
>
> **A3FL under continuous FL poisoning.** A3FL is designed for a late-phase poisoning scenario in which the attacker remains inactive until the main task has largely converged. To evaluate A3FL **fairly and compatibly** under our **continuous-poisoning threat model**, we adopt the **minimal modification needed for applicability**. All clients behave benignly for the first 100 rounds. At round 100, a single malicious client optimizes the trigger using A3FL and shares the resulting trigger with all other malicious clients. Once the trigger is fixed, all malicious clients initiate continuous poisoning for the remaining 200 rounds (as described in Sec. 4.1). In contrast, the original A3FL assumes that attacks are launched only during the last 100 rounds with random frequency. Therefore, this results in a substantially stronger attack compared to the original A3FL formulation. For AUPF, we also examine a variant in which the malicious clients perform standard poisoning training, without the adaptive attack described in Appendix B.1.
>
> **Baseline defenses fail under the strengthened A3FL attack.** Under this strengthened A3FL attack, existing defenses significantly degrade. The results in terms of ACC/ASR (%) on CIFAR-10 are summarized below. For Lockdown, the default configuration **fails completely (ASR ≈ 100%)**. We therefore increase its defense budget (anneal_ratio from 0.0001 to 0.1), but even with this strengthened configuration, the ASR only drops to **77%** and the ACC decreases to **71%**. Other defenses (FDCR, Snowball) also completely fail, all producing **ASR above 99%**. In contrast, **AUPF remains highly robust under A3FL.**
>
> ### ACC / ASR (%) of different defenses on CIFAR-10 under the A3FL attack
>
> | FedAvg        | FDCR          | Snowball      | Lockdown (increased budget) | **AUPF**        | **AUPF (no adaptive attack)** |
> |---------------|---------------|---------------|------------------------------|------------------|-------------------------------|
> | 88.63 / 99.99 | 86.85 / 99.77 | 82.33 / 100.00 | 71.40 / 77.14               | **80.34 / 36.18** | **84.39 / 22.40**             |
>
> These results show that: 1) **AUPF substantially outperforms all existing defenses under A3FL**, 2) **AUPF remains robust even against an unlearning-targeted adaptive attack**, and 3) the non-adaptive attacker variant provides the **upper bound of AUPF** when the attacker does not perform adversarial training.
>
>
>
> **Chameleon** utilizes contrastive learning to amplify the backdoor effect, thereby enhancing its **long-term persistence**. We evaluate Chameleon to gain valuable insights into AUPF’s robustness against stealthy, persistence-oriented attacks. In this experiment, we set the malicious client ratio to 10% (compared to 20% in our paper), as a 20% ratio would lead to excessively low ACC in this setting. **All other experimental settings follow Sec. 4.1 of our paper.** To ensure comparability, we follow the **original Chameleon attack schedule**: 1) The adversary poisons during **rounds 200 to 300**; 2) The adversary remains inactive during **rounds 300 to 800**. This setup is specifically designed to evaluate whether the backdoor can persist long after poisoning has stopped.
>
>
> The results are provided in Fig.~S7 on page~20 of the revised submission. As shown in the results, **FedAvg exhibits strong long-term vulnerability.** Consistent with the original Chameleon paper, FedAvg maintains a high ASR long after the attacker becomes inactive. With only 100 rounds of poisoning (10% malicious clients, pixel “plus” trigger), FedAvg’s ASR remains around **75% at round 800**, indicating significant persistence of the backdoor.
>
> **AUPF breaks Chameleon’s persistence mechanism.** In contrast, AUPF prevents the backdoor from persisting. ASR shows a brief increase during the poisoning window but steadily diminishes once the attack ends. This suggests that AUPF prevents the attacker from embedding stable, convergence-resistant backdoor directions in the feature space, removing the durability advantage characteristic of Chameleon.
>
> **ACC stability.** The brief fluctuation in ACC around round 200 is expected, as Chameleon’s poisoned optimization dynamics differ from benign training under our *from scratch* setup. After this transient period, AUPF stabilizes and maintains accuracy for the remainder of training.
>
> Overall, these results indicate that **FedAvg is highly vulnerable to persistent backdoor attacks**, while **AUPF suppresses the attacker’s ability to maintain long-lasting backdoor effects**.

---

> > ### Author Response · Authors · 2025-11-20
> >
> > **Weakness 3: Missing evaluation against recent defenses specifically designed for adaptive attacks, such as AlignIns.**
> >
> > **Response**: We thank the reviewer for mentioning AlignIns. We first compare AlignIns and AUPF on CIFAR-10 under the same settings as Table 1 of our submission. While AlignIns maintains relatively high clean accuracy, it **fails to suppress ASR** across all four attacks.
> >
> > ### Performance comparison (ACC / ASR %) between AlignIns and AUPF on CIFAR-10
> >
> > | **Attack**     | **BadNet**     | **DBA**        | **Neurotoxin** | **LP**          |
> > |----------------|----------------|----------------|----------------|-----------------|
> > | **AlignIns**   | 87.93 / 99.34  | 88.77 / 88.47  | 87.13 / 99.75  | 88.73 / 88.98   |
> > | **AUPF**       | **84.61 / 4.96** | **85.66 / 2.03** | **84.49 / 4.80** | **84.97 / 8.00** |
> >
> > As discussed in Appendix A.3, AlignIns is a **server-side robust aggregation-based defense** that detects malicious client updates prior to aggregation. Using the same analysis protocol outlined in Appendix A.3, we evaluated AlignIns under varying levels of data heterogeneity and different poison ratios, both of which are critical factors that influence the performance of robust aggregation methods. Specifically, we consider four settings: 1) IID, poison ratio = 0.5; 2) IID, poison ratio = 0.2; 3) non-IID, poison ratio = 0.5; 4) non-IID, poison ratio = 0.2. Using BadNet on CIFAR-10, we obtain the following results:
> >
> > ### FNR and ASR (%) of AlignIns under different data heterogeneity and poison ratios (CIFAR-10, BadNet)
> > | **Setting**           | **IID, PR=0.5** | **IID, PR=0.2** | **non-IID, PR=0.5** | **non-IID, PR=0.2** |
> > |-----------------------|------------------|------------------|-----------------------|-----------------------|
> > | **FNR (%)**           | 0.00             | 0.17             | 12.50                 | 29.17                 |
> > | **ASR (%)**           | 0.81             | 0.75             | 78.40                 | 99.34                 |
> >
> > These results demonstrate that AlignIns performs well when the data are IID and the poison ratio is high, as malicious updates can be more easily detected in such scenarios. However, its detection ability rapidly degrades when the **data become non-IID** and the **poison ratio decreases**, making benign and malicious updates much harder to distinguish. This corresponds exactly to realistic FL scenarios in which robust aggregation methods typically break down.
> >
> > Overall, the results confirm that while AlignIns is effective at detecting malicious updates in ideal scenarios, **it struggles when the poison ratio is small and the data distribution is non-IID**. In contrast, AUPF consistently achieves low ASR across all attacks by performing **local adversarial unlearning** on benign clients, which does not rely on separability between benign and malicious updates. We will include a short discussion in the revision to clarify these differences.

---

> ### Author Response · Authors · 2025-11-24
>
> Dear Reviewer ppHF,
>
> Thank you again for your thoughtful and constructive comments, as well as your positive assessment of the strengths of our work.
>
> We have carefully addressed all the points you raised in our rebuttal, including additional analysis, experiments, and clarifications. Whenever convenient, we would be grateful if you could take a moment to see whether our responses sufficiently clarify the issues you highlighted.
>
> Please feel free to let us know if any part would benefit from further explanation. We are more than happy to provide any additional details.
>
> Thank you again for your time and valuable feedback.
>
>
> Sincerely,
>
> Authors

---

### Meta-Review · Area_Chair_Mzyk · 2026-01-08

**Summary:**

Reviewers recognized the importance of the problem and strong empirical results, but questioned the conceptual novelty of the approach and the lack of theoretical support for the claimed unlearning behavior. They also raised concerns about the completeness of the evaluation, including coverage of adaptive attacks, baselines, attacker-ratio sensitivity, and practical deployment considerations. The rebuttal addressed many of these points through additional experiments and clarifications, improving confidence in the correctness and robustness of the empirical claims, but did not fully resolved the novelty and contribution-level concerns. These suggest that the work may benefit from further development before being competitive for acceptance at ICLR.

**Reviewer Concerns:**

Concerns largely addressed:
- The rebuttal clearly clarifies unlearning objective and provides mechanistic evidence via t-SNE visualizations and ablation studies showing the necessity of the unlearning term. Additional experiments with A3FL, Chameleon, and AlignIns directly address several claims about weak adaptive evaluation and missing state-of-the-art baselines. The rebuttal clarifies partial participation and attacker-ratio sensitivity. New results on ResNet-50 and larger client populations address concerns about computational and cross-device scalability, and clarify that communication overhead.

Outstanding Concerns:
- Even after clarification, it remains debatable whether the contribution rises above a careful integration of known components into a clearly distinct conceptual advance.  Questions about the generality of the approach, the strength of the evaluation protocol, and the absence of rigorous theoretical or empirical justification were not adequately addressed.

**Reviewer Scores:**

- Reviewer ppHF (initial score: 6): Likely to remain unchanged.
- Reviewer xjUd (initial score: 4): Likely to remain unchanged.
- Reviewer 2GMK (initial score: 4): Likely to remain unchanged.

---

### Decision · Program_Chairs · 2026-01-26

Reject